# UniDSeg: Unified Cross-Domain 3D Semantic Segmentation via Visual Foundation Models Prior

**Yao Wu**[1], **Mingwei Xing**[2], **Yachao Zhang**[1]*, **Xiaotong Luo**[1], **Yuan Xie**[4,5], **Yanyun Qu**[1,2,3]*
[1]School of Informatics, Xiamen University
[2]Institute of Artificial Intelligence, Xiamen University
[3]Key Laboratory of Multimedia Trusted Perception and Efficient Computing,
Ministry of Education of China, Xiamen University
[4]School of Computer Science and Technology, East China Normal University
[5]Chongqing Institute of East China Normal University
`wuyao@stu.xmu.edu.cn, yyqu@xmu.edu.cn`

## Abstract

3D semantic segmentation using an adapting model trained from a source domain with or without accessing unlabeled target-domain data is the fundamental task in computer vision, containing domain adaptation and domain generalization. The essence of simultaneously solving cross-domain tasks is to enhance the generalizability of the encoder. In light of this, we propose a groundbreaking universal method with the help of off-the-shelf Visual Foundation Models (VFMs) to boost the adaptability and generalizability of cross-domain 3D semantic segmentation, dubbed **UniDSeg**. Our method explores the VFMs prior and how to harness them, aiming to inherit the recognition ability of VFMs. Specifically, this method introduces layer-wise learnable blocks to the VFMs, which hinges on alternately learning two representations during training: *(i)* Learning visual prompt. The 3D-to-2D transitional prior and task-shared knowledge is captured from the prompt space, and then *(ii)* Learning deep query. Spatial Tunability is constructed to the representation of distinct instances driven by prompts in the query space. Integrating these representations into a cross-modal learning framework, UniDSeg efficiently mitigates the domain gap between 2D and 3D modalities, achieving unified cross-domain 3D semantic segmentation. Extensive experiments demonstrate the effectiveness of our method across widely recognized tasks and datasets, all achieving superior performance over state-of-the-art methods. Remarkably, UniDSeg achieves 57.5%/54.4% mIoU on "A2D2/sKITTI" for domain adaptive/generalized tasks. Code is available at `https://github.com/Barcaaaa/UniDSeg`.

## 1 Introduction

As deep learning technology develops by leaps and bounds [47, 57, 59], 3D scene understanding has become the foundation for many real-world applications, including autonomous driving, robotics, augmented reality, smart cities, etc. Based on the LiDAR sensor, 3D semantic segmentation is a critical task that provides an accurate and robust semantic prediction of the surrounding scenarios. However, annotating large-scale datasets for training every new scenario is labor-intensive and time-consuming, especially for the tasks demanding point-wise annotations. These limitations hinder their practical applicability in real-world scenarios where acquiring high-quality labeled data.

Currently, domain adaptive 3D semantic segmentation (DA3SS) [19, 32, 38, 45, 48, 51] and domain generalized 3D semantic segmentation (DG3SS) [21, 27, 49, 60] have been widely explored in

---

*Corresponding author

38th Conference on Neural Information Processing Systems (NeurIPS 2024).

autonomous driving scenes. Their difference lies in that the former seeks to narrow the domain gap between the source and target domain data without assigning 3D semantic labels, while the latter aims to learn a generic and robust model before being exposed to the target domain. Albeit successful, their applications in 3D semantic segmentation have primarily focused on generalizing or adapting between synthetic and real scenes or across different scene layouts. This leaves a gap in exploring a universal framework, enabling the generalization and adaptation of 3SS models across datasets.

With the above considerations, this paper focuses on studying a universal framework for cross-domain 3D semantic segmentation. The essence of simultaneously solving cross-domain tasks is to enhance the generalizability of the encoder. Therefore, a generalizable 3D model with source-domain data discrimination power to the target domain is necessary. It is performed solely with access to source domain data, enabling the model to develop the ability to discriminate domain-agnostic and domain-specific features. Unfortunately, one limitation has arisen, the scarcity of 3D pre-training datasets hinders this endeavor. More recently, Visual Foundation Models (VFMs) [23, 34, 36] have emerged as the de-facto visual backbone in 2D image classification and segmentation. Such VFMs are trained on massive raw web-curated images, achieving promising open-vocabulary recognition. Hence, two natural questions are thrown up: *(i) How to borrow 2D prior knowledge from VFMs?* and *(ii) How to harness 2D prior knowledge to boost 3D performance?* To begin with, for issue *(i)*, visual prompt tuning [16, 20] is a parameter-efficient strategy to exploit the representational potential of VFMs. We consider freezing the whole VFMs and only learn several trainable lightweight blocks as supplementary input, which inherit parameters of VFMs trained at scale to the maximum extent. After that, for issue *(ii)*, we consider a more effective prompt tuning that introduces an extra depth-guided prompt space and extends the model input with the prompt, which could guide the generalization of powerful representations to achieve desirable performances.

Accordingly, in this paper, we introduce **UniDSeg**, dig deeper into prompt tuning in the VFM-based encoder, and introduce a *Learnable-parameter-inspired Mechanism* to the off-the-shelf VFMs with frozen parameters. Our VFM-based encoder is designed to learn alternately between two lightweight modules: *i.e.,* Modal Transitional Prompting (MTP) and Learnable Spatial Tunability (LST). The former depends on the transitional guidance from 3D-to-2D unnatural images, *i.e.,* sparse depth, which exists in the prompt space before being fed into the encoder layer. The latter depends on the customized context length of vectors, which exists in the query space for seeking matched prompting after encoding in the layer. Hereby, depending on the number of VFM-based encoder layers involved, we place layer-wise MTP and LST blocks to take full advantage of semantic understanding of diverse levels and modalities. The proposed mechanism not only avoids any unnecessary attempts to manipulate the original visual space but also inherits the pre-existing target awareness from the VFMs to the maximum extent. Ultimately, by integrating the proposed two modules into a cross-modal learning framework, our method efficiently mitigates the domain gap and enables 2D and 3D models to learn domain-invariant representations.

The key contributions of our work are summarized as follows: 1) Our method is groundbreaking in introducing the prompt-tuning concept into the universal model for DG3SS and DA3SS tasks. 2) We propose a novel learnable-parameter-inspired mechanism to the off-the-shelf VFMs, which maximally preserves pre-existing target awareness in VFMs to further enhance its generalizability. 3) Extensive experimental results demonstrate the effectiveness of our method across widely recognized tasks and datasets, all achieving superior performance for DG3SS and DA3SS.

## 2   Related Works

### 2.1   Domain Adaptive 3D Semantic Segmentation

In general, DA3SS seeks to narrow the domain gap between the source and target domain data, which can be grouped as uni-modal [61, 54, 48, 38, 56, 55, 24, 32] and multi-modal [19, 30, 35, 28, 58, 50, 45, 6, 5, 51, 13, 46] conditions. For uni-modality, early methods [48, 61] exploit the generative adversarial network to mitigate domain shift caused by appearance and sparsity differences. Later on, Yuan *et al.* [55, 56] propose the adversarial network based on category-level and prototype-level alignments to address the mismatch of sampling patterns. CosMix [38] and ConDA [24] construct an intermediate domain by utilizing joint supervision signals from both the source and target domains for self-training. 3D surface representation is also considered an effective method. Complete&Label [54]

transforms domain adaptive task into a 3D surface completion task. SALUDA [32] learns an implicit underlying surface representation simultaneously on source and target data.

Compared to uni-modality, multi-modality exploits the exclusive information of paired images and point clouds to complement each other. xMUDA [19] is a pioneering method of cross-modal mutual learning for DA3SS. To facilitate learning domain-robust dependencies, several methods extend 2D techniques to learn the 3D domain-invariant representations, such as adversarial learning [30, 35, 58], style transfer [28], and contrastive learning [50]. Based on these dependencies, SSE [58] presents a self-supervised learning mechanism from plane-to-spatial and discrete-to-textured representations. BFtD [45] presents cross-modal fusion-then-distillation to mitigate imbalanced modality adaptability. Recently, Segment Anything Model (SAM) [23] has presented its strong capability in generating semantics-aware regions, making it a suitable option for cross-modal interaction. MoPA [5] and VFMSeg [51] harness the knowledge priors learned from SAM to produce more accurate pseudo-labels for unlabeled target domains.

In contrast to DA3SS, where the inputs in the target domain, although without 3D labels, are accessible during the training process, DG3SS is evaluated on data from totally unseen target domains.

## 2.2 Domain Generalized 3D Semantic Segmentation

The goal of DG3SS is to first learn as generic and robust representations as possible before being exposed to any target-domain data during training. Research on DG3SS has recently witnessed a surge, as highlighted in several studies, including uni-modal condition [21, 37, 39, 40, 49, 60] and multi-modal condition [27, 14]. Kim *et al.* [21] leverage sparsity invariant feature consistency at the feature level and semantic correlation consistency at the output level to constrain the model. Ryu *et al.* [37] and Sanchez *et al.* [40] incorporate multi-frame aggregation with 6-DoF ego-motions via randomized LiDAR configurations augmentation and label propagation, respectively. However, these methods are limited when 6-DoF ego-motion is unknown, and they need to be estimated using the off-the-shelf LiDAR SLAM method [2]. Recently, 3D representation under Bird's-Eye-View (BEV) has also been considered an effective method for learning domain-invariant features. LiDOG [39] introduces a dense top-down prediction auxiliary task and supervises it by employing BEV-view of semantic labels, while BEV-DG [27] introduces a BEV-view of cross-modal representation fusion to alleviate the domain gap. Particularly, 3D scenes also exist in several adverse weather conditions including fog, snow, and rain [49]. UniMix [60] leverages physically valid adverse weather simulation to construct a bridge domain and then blends it with samples of normal weather conditions.

Our endeavor is tailored to designing a universal cross-domain multi-modal learning framework that enhances the performance of both DG3SS and DA3SS.

## 3 Method

### 3.1 Preliminary

**Problem Definition.** Given a source domain $\mathcal{D}_S = \{(X_i^{2D,S}, X_i^{3D,S}, Y_i^{3D,S})\}_{i=1}^{n_s}$ with $n_s$ labeled data and a target domain $\mathcal{D}_T = \{(X_i^{2D,T}, X_i^{3D,T})\}_{i=1}^{n_t}$ with $n_t$ unlabeled data under the condition that the source and target data distributions are not equal. For DA3SS, the task seeks to adapt models trained on the source domain $\mathcal{D}_S$ to a target domain $\mathcal{D}_T$ without labels. For DG3SS, the task aims to exploit the knowledge from the source domain to achieve generalization to the target domain, while the model remains unexposed to the target domain during training, *i.e.*, $\mathcal{D}_T$ is unseen. Both tasks are framed with the expectation of having paired images and point clouds and learning a mapping function $f : X^{2D,T}, X^{3D,T} \rightarrow Y^{3D,T}$ that could predict the target-domain 3D labels.

**Revisiting ViT.** Given a pre-trained Vision Transformer (ViT) [10] model with $L$ layers, an input is divided into $M = \frac{HW}{P^2}$ fixed-size patches $x_m \in \mathbb{R}^{\frac{H}{P} \times \frac{W}{P} \times D}, m = 1, 2, ..., M$, where $H$ and $W$ are the height and width of the original input, $P$ is the patch size, and $D$ is the constant latent vector size through all of its layers. Each patch is then embedded into $D$-dimensional latent space:

$$e_0^m = Embed(x_m), \quad e_0^m \in \mathbb{R}^D. \tag{1}$$

We indicate the collection of 2D patch embeddings, $E_{l-1} = \{e_{l-1}^m \in \mathbb{R}^D | 1 \leq m \leq M\}$, as input to the $l$-th ViT encoder layer $L_l(\cdot)$. Specially, $E_0^{pre} \in \mathbb{R}^{M \times D}$ learns beforehand together with class

encoding $E_{cls} \in \mathbb{R}^{1 \times D}$ and positional encoding $E_{pos} \in \mathbb{R}^{(1+M) \times D}$ to create initial patch embedding $E_0$. Hence, the whole ViT encoder is formulated as:

$$E_0 = (E_{cls} \uplus E_0^{pre}) + E_{pos}, \tag{2}$$

$$E_l = L_l(E_{l-1}), \quad l = 1, 2, ..., L, \tag{3}$$

where $E_l \in \mathbb{R}^{(1+M) \times D}$ is the patch embedding output from $L_l(\cdot)$, $\uplus$ denotes concatenation on the sequence length dimension. Each layer $L_l$ consists of several multi-head self-attention modules and feed-forward networks together with skip connection [17] and LayerNorm [1].

### 3.2 Learnable-parameter-inspired Mechanism

**Overall Framework.** As depicted in Fig. 1, the overall framework of UniD-Seg is decomposed to a 2D network $\mathcal{F}^{2D} = \mathcal{E}^{2D} \circ \mathcal{G}^{2D} \circ \mathcal{H}^{2D}$ and a 3D network $\mathcal{F}^{3D} = \mathcal{E}^{3D} \circ \mathcal{G}^{3D} \circ \mathcal{H}^{3D}$, where $\mathcal{E}^{(\cdot)}$, $\mathcal{G}^{(\cdot)}$, and $\mathcal{H}^{(\cdot)}$ denote encoder, decoder, and classifier, respectively. The main insight of UniDSeg is to provide a universal framework that enhances the generalizability and adaptability of cross-domain 3D semantic segmentation. Thereby, we introduce a novel task-specific VFM-based encoder, which is guided by point-level prompts from 3D information. Formally, point cloud $X_i^{3D}$ is input to 3D network $\mathcal{F}^{3D}$ to generate 3D prediction, while image $X_i^{2D}$ and sparse depth $X_i^{Dep}$ are input to 2D network $\mathcal{F}^{2D}$ to generate 2D prediction with the help of VFMs priors. $X_i^{Dep}$ is derived from the LiDAR sensor via per-

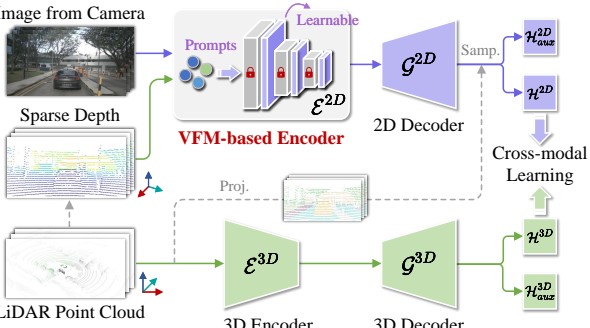

Figure 1: Overall framework of UniDSeg for DG3SS and DA3SS. The backbone of the VFM-based Encoder is frozen and only trains several learnable modules. "Samp." means sampling of 2D features. Only the points falling into the intersected field of view are geometrically associated with multi-modal data.

spective projection. Afterward, we employ cross-modal learning on the source/target predictions via auxiliary classifier $\mathcal{H}_{aux}^{(\cdot)}$. In summary, we place layer-wise learnable blocks to take full advantage of semantic understanding of diverse levels and modalities, which inherits potential target information of VFMs into the current training model. Our method is parameter-efficient and could be directly deployed to various pre-trained transformer-based architectures without modifying the basic units.

**VFM-based Encoder.** Since the feature extraction of VFMs still lacks task-specific information, freezing parameters may lose some domain-agnostic contextual semantic information from source-domain data or fine-tuning parameters may alter some pre-existing target representation from large-scale pre-trained data. To provide a remedy, we reconsider the role of pre-trained models, balancing the capture of contextual information and the overfitting issue of source-domain data. As illustrated in Fig. 2, we introduce a *Learnable-parameter-inspired Mechanism*, which provides a set of continuous embedding, *i.e.,* 3D-to-2D transitional prompts and tunable deep queries. The former not only learns spatial distance perception prompts from point clouds but also learns invariance to sample perturbations. The latter ensures that the feature distribution of $\tilde{E}_l$ will not be modified drastically, thus making better use of the pre-trained knowledge from VFMs. Only these prompts and queries are updated during fine-tuning, while parameters of other layers $L_l$ of the ViT encoder are kept frozen.

Depending on the number of ViT layers involved, our VFM-based encoder is designed to learn alternately between two lightweight modules: *Modal Transitional Prompting* $PG_l(\cdot, \cdot)$ and *Learnable Spatial Tunability* $TB_l(\cdot)$. The former depends on the transitional guidance from 3D-to-2D unnatural images, which exists in the prompt space before being fed into layer $L_l$, while the latter depends on the customized context length of vectors, which exists in the query space for seeking matched prompting after encoding in layer $L_l$. With deliberate design, the newer task-specific ViT encoder can be decomposed into:

$$\tilde{E}_l = L_l(\tilde{E}_{l-1}) + TB_l(L_l(\tilde{E}_{l-1})), \quad l = 1, 2, ..., L, \tag{4}$$

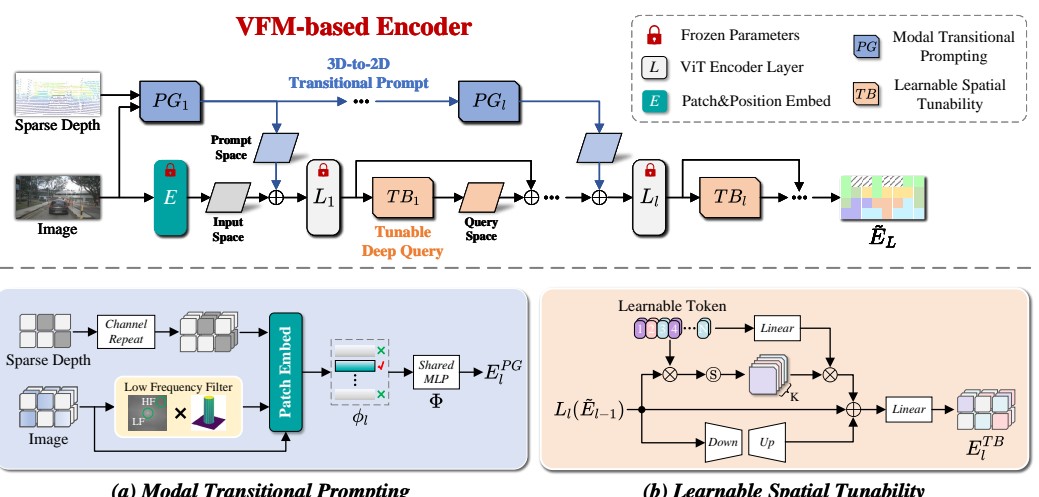

(a) *Modal Transitional Prompting*      (b) *Learnable Spatial Tunability*

Figure 2: The architecture of VFM-based Encoder. We explore two layer-wise learnable blocks: (a) Modal Transitional Prompting and (b) Learnable Spatial Tunability. During training, only the parameters of two modules are updated while the whole ViT encoder layer is frozen.

$$\tilde{E}_{l-1}[1:,:] = E_{l-1}[1:,:] + PG_l(X^{2D}, X^{Dep}), \tag{5}$$

where $\tilde{E}_L$ is the final refined patch embedding output from our proposed VFM-based encoder. According to the DG or DA task, $X^{2D} \in \mathbb{R}^{H \times W \times 3}$ means source or target image and $X^{Dep} \in \mathbb{R}^{H \times W \times 1}$ means source or target sparse depth. Note that as Eq. (5) shows, the prompt output from $PG_l$ is aligned to the patch embedding without contacting $E_{cls}$.

**Modal Transitional Prompting.** Previous methods [16, 20] have proved that visual prompt-tuning brings flexibility to the pre-trained VFMs for downstream tasks. However, their prompts like pixel-level perturbations or learnable vectors are black boxes with limited learning capacity, which cannot reliably explore convincing knowledge beneficial for cross-domain 3D semantic segmentation. Lee *et al.* [26] have proved that source data internally know much more about the world and how the scene is formed, called Privileged Information. Therefore, $PG_l$ is designed to capture 3D-to-2D transitional prior and task-shared knowledge of this information from the prompt space, which might be useful for cross-domain learning.

To achieve this goal, following Eq. (1), a patch embed module borrowed from the plain ViT is re-initialized and applied to obtain a sequence of flattened patch embeddings $E_0^{2D}, E_0^{Dep} \in \mathbb{R}^{M \times D}$. Particularly, sparse depth $X^{Dep}$ as a point cloud representation via perspective projection, presents an unnatural image. From the view of modal characteristics, it is easy to access and contains the prior spatial distance information that is lacking in 2D representation. From the view of deep encoding, it focuses on the scope of scenes at different receptive fields, tightly coupled with semantic information, so that their corresponding features have different contents when constructed. Note that $X^{Dep}$ only has the 1-channel to show the depth value in camera coordinates, we repeat the depth channel to make it equal to the number of RGB channels. In addition, considering that low-frequency bands of the amplitude spectrum tend to capture style information (or low-level statistics) [53], we provide a simple way to perturb the amplitude spectra of the source image to ensure that the VFM is exposed to more variations in low-frequency components during training. The perturbed images can be obtained by using a low-frequency filter, and then Eq. (1) is also employed to generate corresponding flattened patch embedding $E_0^{LF} \in \mathbb{R}^{M \times D}$. After that, to effectively integrate prompt-tuning to visual embedding output from each frozen layer, we flexibly build a lightweight layer that contains exclusive MLP $\phi_l(\cdot)$ for adapting $l$-th encoder layer and shared MLP $\Phi(\cdot)$ across whole ViT. This implementation is written as:

$$E_l^{PG} = \Phi(\phi_l(E_0^{2D} \uplus E_0^{LF} \uplus E_0^{Dep})), \tag{6}$$

where $E_l^{PG} \in \mathbb{R}^{M \times D}$ is the 3D-to-2D transitional prompt output from $PG_l$. MLP is built with a two-layer bottleneck structure (Linear-GELU-Linear) with the hidden layer reducing the input dimension by $3\times$.

**Learnable Spatial Tunability.** To progressively improve feature generalization via 3D-to-2D transitional prompt, inspired by [31, 44], $TB_l$ is introduced to bridge the discrepancy between the pre-training dataset and the target scene. To achieve this goal, after the $l$-th layer, $TB_l$ starts with a set of learnable tokens $O_l \in \mathbb{R}^{K \times D}$, where $K$ is the length of the token and each token is initialized randomly. Considering the essential need in 3D semantic segmentation to discern multiple instances within a framed scene, $TB_l$ exploits an attention mechanism, which enables VFMs to seek matched prompting to the features of distinct instances, thereby assisting VFMs in adapting to the differences between pre-training and cross-domain data. Concretely, $TB_l$ adopts a dot-product operation to generate the affinity matrix $J_l$, which captures the associations between prompted patch embedding $L_l(\tilde{E}_{l-1})$ and learnable tokens $O_l$. Then, a *SoftMax* function is applied to align each patch with a unique instance. This implementation is written as:

$$J_l = SoftMax(\frac{L_l(\tilde{E}_{l-1})[1:,:] \times O_l^\top}{\sqrt{D}}), \quad J_l \in \mathbb{R}^{M \times K}, \tag{7}$$

$$O_l = O_{l,a} \times O_{l,b}, \tag{8}$$

where $O_{l,a} \in \mathbb{R}^{K \times D_m}$ and $O_{l,b} \in \mathbb{R}^{D_m \times D}$ are constructed as low-rank matrices [18], reducing the trainable parameters by learning pairs of rank-decomposition matrices. Note that the low-rank dimension $D_m \ll D$ ($D_m = 32$ in our case). Besides, we further process the embeddings through a down-projection layer with parameters $W_{down} \in \mathbb{R}^{K \times D_m}$ followed by an up-projection layer with parameters $W_{up} \in \mathbb{R}^{D_m \times D}$, which is then element-wise added via the residual connection. After that, to enhance the flexibility in feature adjustment, $TB_l$ employs several layers to produce:

$$E_l^{TB} = \delta_2(L_l(\tilde{E}_{l-1})[1:,:] + W_{up}^\top \times (W_{down}^\top \times L_l(\tilde{E}_{l-1})[1:,:]) + J_l \times \delta_1(O_l)), \tag{9}$$

where $E_l^{TB} \in \mathbb{R}^{M \times D}$ is the tunable deep query output from $TB_l$, $\delta_1(\cdot)$ and $\delta_2(\cdot)$ are linear layers.

## 3.3 Cross-modal Learning

The point-wise supervised segmentation loss of the source domain is formulated as follows:

$$\mathcal{L}_{seg} = -\frac{1}{N \times C} \sum_{n=1}^{N} \sum_{c=1}^{C} Y_{(n,c)}^{3D,S} \log \mathbf{P}_{(n,c)}^S, \tag{10}$$

where main prediction $\mathbf{P}^S$ is either $\mathbf{P}^{2D,S}$ or $\mathbf{P}^{3D,S}$, $N$ and $C$ being the number of points of the source point cloud and the number of classes, respectively.

The goal of unsupervised learning across modalities is two-fold. Firstly, we want to transfer knowledge from 2D modality to 3D modality on the source-domain and target-domain dataset. Secondly, we devise mutual learning on the output level, where the task is to transfer the pre-existing target information of VFMs to a 3D model. Same to xMUDA [19], we choose the Kullback-Leibler divergence $D_{KL}(\cdot||\cdot)$ for the cross-modal loss $\mathcal{L}_{xM}$ and define it as follows:

$$\mathcal{L}_{xM}^S = D_{KL}(\mathbf{P}^{2D,S}||\mathbf{P}^{3D,S\mapsto2D,S}) + D_{KL}(\mathbf{P}^{3D,S}||\mathbf{P}^{2D,S\mapsto3D,S}), \tag{11}$$

$$\mathcal{L}_{xM}^T = D_{KL}(\mathbf{P}^{2D,T}||\mathbf{P}^{3D,T\mapsto2D,T}) + D_{KL}(\mathbf{P}^{3D,T}||\mathbf{P}^{2D,T\mapsto3D,T}), \tag{12}$$

where $\mathbf{P}^{2D,S}$ and $\mathbf{P}^{3D,S}$ is to be estimated by mimicking predictions $\mathbf{P}^{3D,S\mapsto2D,S}$ and $\mathbf{P}^{2D,S\mapsto3D,S}$ from the respective auxiliary classifiers $\mathcal{H}_{aux}^{2D}$ and $\mathcal{H}_{aux}^{3D}$. The same goes for target predictions. Ultimately, the overall loss function $\mathcal{L}_{DG}$ of DG3SS and $\mathcal{L}_{DA}$ of DA3SS are defined as:

$$\mathcal{L}_{DG} = \mathcal{L}_{seg} + \lambda_S \mathcal{L}_{xM}^S, \tag{13}$$

$$\mathcal{L}_{DA} = \mathcal{L}_{seg} + \lambda_S \mathcal{L}_{xM}^S + \lambda_T \mathcal{L}_{xM}^T, \tag{14}$$

where $\lambda_S$ and $\lambda_T$ are the weights trading off cross-modal loss on source and target domain inputs.

# 4 Experiments

## 4.1 Datasets

For evaluation, we use four public autonomous driving benchmarks, including three real datasets: *nuScenes* [4], *SemanticKITTI* [3], *A2D2* [12] and one synthetic dataset: *VirtualKITTI* [11]. For all

Table 1: Performance comparison of multi-modal domain adaptive and domain generalized 3D semantic segmentation methods in four typical scenarios. We report the mIoU results (with **best** and 2nd best) on the target testing set for each network as well as the ensemble result (*i.e.*, xM) by averaging the predicted probabilities from the 2D and 3D networks.

| S:Source / T:Target | | nuScenes:USA/Sing. | | | nuScenes:Day/Night | | | vKITTI/sKITTI | | | A2D2/sKITTI | | |
|---|---|---|---|---|---|---|---|---|---|---|---|---|---|
| Task | Method | 2D | 3D | xM | 2D | 3D | xM | 2D | 3D | xM | 2D | 3D | xM |
| | Source-only | 58.4 | 62.8 | 68.2 | 47.8 | 68.8 | 63.3 | 26.8 | 42.0 | 42.2 | 34.2 | 35.9 | 40.4 |
| | logCORAL [33] | 64.4 | 63.2 | 69.4 | 47.7 | 68.7 | 63.7 | 41.4 | 36.8 | 47.0 | 35.1 | 41.0 | 42.2 |
| | MinEnt [43] | 57.6 | 61.5 | 66.0 | 47.1 | 68.8 | 63.6 | 39.2 | 43.3 | 47.1 | 37.8 | 39.6 | 42.6 |
| | BDL [29] | 62.0 | 64.8 | 70.4 | 47.0 | 69.6 | 63.0 | 21.5 | 44.3 | 35.6 | 34.7 | 41.7 | 45.2 |
| DA | xMUDA [19] | 64.4 | 63.2 | 69.4 | 55.5 | 69.2 | 67.4 | 42.1 | 46.7 | 48.2 | 38.3 | 46.0 | 44.0 |
| | AUDA [30] | 64.0 | 64.0 | 69.2 | 55.6 | 69.8 | 64.8 | 35.8 | 37.8 | 41.3 | 43.0 | 43.6 | 46.8 |
| | DsCML [35] | 65.6 | 56.2 | 66.1 | 50.9 | 49.3 | 53.2 | 38.4 | 38.4 | 45.5 | 39.6 | 45.1 | 44.5 |
| | Dual-Cross [28] | 64.7 | 58.1 | 66.5 | 58.5 | 69.7 | 68.0 | 40.7 | 35.1 | 44.2 | 45.9 | 40.0 | 48.6 |
| | SSE [58] | 64.9 | 63.9 | 69.2 | 62.8 | 69.0 | 68.9 | 45.9 | 40.0 | 49.6 | 44.5 | 46.8 | 48.4 |
| | BFtD [45] | 63.7 | 62.2 | 69.4 | 57.1 | 70.4 | 68.3 | 41.5 | 45.5 | 51.5 | 40.5 | 44.4 | 48.7 |
| | MM2D3D [6] | 71.7 | 66.8 | 72.4 | 70.5 | 70.2 | **72.1** | 53.4 | 50.3 | 56.5 | 42.3 | 46.1 | 46.2 |
| | VFMSeg [51] | 70.0 | 65.6 | 72.3 | 60.6 | 70.5 | 66.5 | 57.2 | 52.0 | 61.0 | 45.0 | 52.3 | 50.0 |
| | **UniDSeg** | 67.2 | 67.6 | **72.9** | 63.2 | 71.2 | 71.2 | 60.5 | 50.9 | **62.0** | 50.7 | 55.4 | **57.5** |
| DG | xMUDA [19] | 58.7 | 62.3 | 68.6 | 43.0 | 68.9 | 59.6 | 25.7 | 37.4 | 39.0 | 34.9 | 36.7 | 41.6 |
| | MM2D3D [6] | 69.7 | 62.3 | 70.9 | 65.3 | 63.2 | 68.3 | 37.7 | 40.2 | 44.2 | 39.6 | 35.9 | 43.6 |
| | **UniDSeg** | 66.5 | 64.5 | **72.3** | 57.0 | 70.5 | **70.0** | 57.6 | 44.7 | **60.0** | 48.8 | 46.3 | **54.4** |

real datasets, LiDAR sensor and RGB cameras are synchronized and calibrated, allowing 2D-to-3D projection, and for the synthetic dataset, *VirtualKITTI* provides depth maps so we simulate LiDAR scanning via uniform point sampling. Following DA3SS settings [19], we exclusively utilize the front camera image and the corresponding LiDAR points that are projected onto it.

Our experimental scenarios cover typical real-to-real single domain adaptation and generalization challenges like lighting changes (*nuScenes*: *Day/Night*), scene layout of country (*nuScenes*: *USA/Sing.*; *nuScenes*: *Sing./USA*), and sensor setups (*A2D2/sKITTI*; *A2D2/nuScenes*). Note that, the source and target classes are coincident. For the first three scenarios, we choose 6 merged classes while for the last two scenarios, we select 10 and 8 shared classes between two datasets. In addition, the synthetic-to-real domain adaptation and generalization challenges are also considered (*vKITTI/sKITTI*, simulated depth, and RGB to real LiDAR and camera, with 6 merged classes). Details of the class partition are provided in supplementary materials.

## 4.2 Implementation Details

For the 2D backbone, we use the visual encoders of CLIP [36] and SAM [23], a large-scale pre-training model. Our selection of Vision Transformer [10] includes ViT-Base (ViT-B) and ViT-Large (ViT-L) architectures. Then, we utilize the MMSegmentation [8] codebase for the decoder head, SemanticFPN [22], a widely-used segmentation head, is integrated with the visual encoder that serves as the 2D backbone. Meanwhile, depth information is a 3D attribute derived from LiDAR sensors, without the need to introduce additional datasets, thus ensuring fairness under equal experimental conditions. For the 3D backbone, we employ SparseConvNet [15] with a U-Net architecture in Sparse Convolution Library [9]. The voxel size is set to 5cm in the 3D network. This voxel size ensures that each voxel contains only one 3D point, maintaining a level of granularity suitable for the task.

Our model is trained on "nuScenes:Day/Night", "A2D2/sKITTI", and "A2D2/nuScenes" for 100k iterations. We utilize an iteration-based learning schedule where the initial learning rate is set to 1e-3 except for the 2D encoder which is 1e-4, and then it is divided by 10 at 80k and 90k iterations. For "nuScenes:USA/Sing." and "nuScenes:Sing./USA", the training is performed for 60k iterations, and the learning rate is divided by 10 at the 40k and 50k iterations. For vKITTI/sKITTI, the training is performed for 30k iterations, and the learning rate is divided by 10 at the 25k and 28k iterations. The batch size is set to 8. As regards the hyper-parameters, following [19], $\lambda_S$ and $\lambda_T$ in cross-modal loss are set to 1.0 and 0.1 on "nuScenes:Day/Night", "nuScenes:USA/Sing.", and "nuScenes:Sing./USA", 0.1 and 0.01 on "vKITTI/sKITTI", "A2D2/sKITTI", and 'A2D2/nuScenes" respectively, without performing any fine-tuning on these values. All experiments are conducted on NVIDIA RTX 3090.

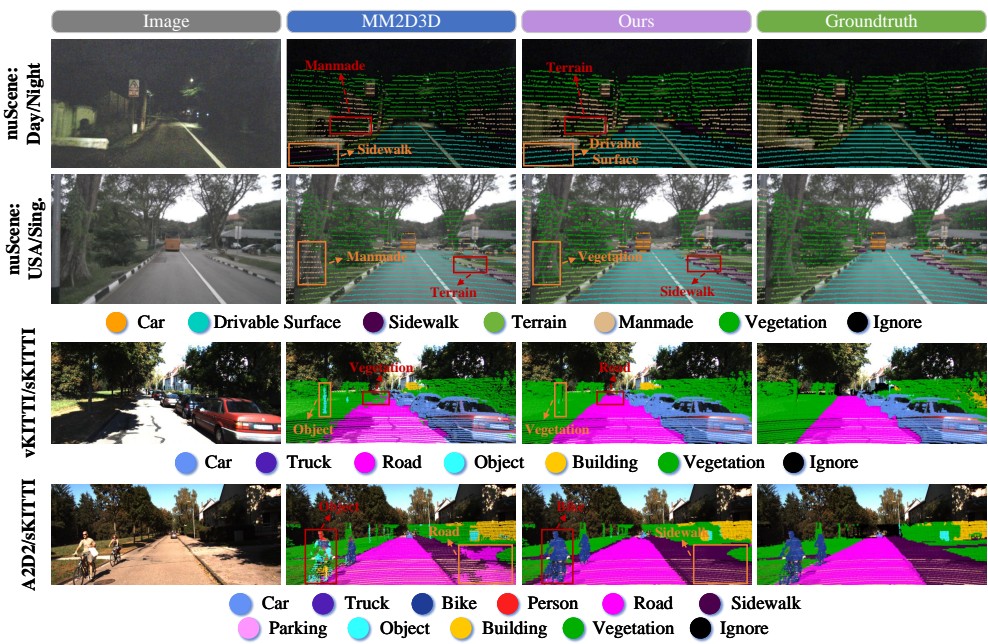

Figure 3: Qualitative results of DG3SS. We showcase the ensembling result of four scenarios.

## 4.3 Quantitative and Qualitative Comparison

We compare the proposed method with three classic 2D domain adaptive methods, which can be easily extended to multi-modal conditions. Moreover, eight multi-modal DA3SS and two multi-modal DG3SS are discussed. As shown in Tab. 1, we tabulate the comparison results in mean Intersection over Union (mIoU, %) on the target testing data. Note that not all datasets provide image labels. Thus the quantitative evaluation of 2D semantic segmentation depends on the 3D-2D corresponding point-wise prediction. Overall, our method achieves the best performance on all scenarios against the competitors, $w.r.t.$ ensemble result "xM", except for DA3SS on "nuScenes:Day/Night" gets the second best. As shown in Fig. 3, we visualize the DG3SS results of four settings. By the merit of the visual foundation models, our method can segment detailed objects very well. From top to bottom, the focused areas are the sidewalk near the drivable surface, the trunk of a tree, the profile of the road, and most importantly, bicycles safely riding on the road.

**Comparison of DG3SS.** In terms of DG task, compared with baseline (xMUDA) and MM2D3D, our method still achieves remarkable results, exhibiting 3.7%/1.4%, 10.4%/1.7%, 21.0%/15.8%, and 12.8%/10.8% mIoU improvement. It is worth noting that our method, without exposure to any target-domain data, shows performance that is close to or even surpasses that of most DA3SS methods (*e.g.*, the result of Ours-DG is comparable to VFMSeg on "nuScenes:USA/Sing."). All results demonstrate that utilizing the powerful open-vocabulary recognition capability of VFMs is beneficial for addressing generalization problems in domains with significant discrepancies.

**Comparison of DA3SS.** The source-only model is the lower bound, which is not domain adaptive as it is only trained on the source-domain data. It is observed that our method brings a significant adaptation effect on all scenarios compared to the source-only model, with the gains of 4.7%, 7.9%, 19.8%, and 17.1% in mIoU, respectively. Compared with baseline (xMUDA) for all DA competitors, our method exceeds it by large margins with gains of 3.5%, 3.8%, 13.8%, and 13.5% in mIoU. Particularly, on the "A2D2/sKITTI" scenario, our method typically yields a higher score compared to the best "xM" in VFMSeg (57.5% vs. 50.0%).

## 4.4 Ablation Study

**Evaluation of Different VFMs Training Strategies.** Our analysis of various training strategies for VFMs in six experimental scenarios within and across datasets can be found in Tab. 2. Note

Table 2: Ablation study on VFM-based encoder with different training strategies for DG3SS. This setup is based on the same LiDAR-Camera configuration but different environments (top-3 scenarios), and different LiDAR-Camera configurations (bottom-3 scenarios). Of note, "Params" denote trainable parameters in the encoder.

| S:Source / T:Target | | | nuScenes:USA/Sing. | | | nuScenes:Day/Night | | | nuScenes:Sing./USA | | |
|---|---|---|---|---|---|---|---|---|---|---|---|
| Strategy | Visual Backbone | Params | 2D | 3D | xM | 2D | 3D | xM | 2D | 3D | xM |
| Finetune | | 86.9M | 62.4 | 64.1 | 69.6 | 53.3 | 70.7 | 68.8 | 65.7 | 67.9 | 72.9 |
| Frozen | CLIP:ViT-B | 0.0M | 59.7 | 64.5 | 69.7 | 46.8 | 71.0 | 69.8 | 58.3 | 67.9 | 71.2 |
| **Ours** | | 1.82M | 63.8 | 64.7 | **71.5** | 55.9 | 70.7 | **70.0** | 68.2 | 68.0 | **74.0** |
| Finetune | | 305M | 65.5 | 64.5 | 70.4 | 54.9 | 70.7 | 67.3 | 69.9 | 67.8 | 74.5 |
| Frozen | CLIP:ViT-L | 0.0M | 60.4 | 64.2 | 70.1 | 50.2 | 70.5 | 69.5 | 62.2 | 67.8 | 73.3 |
| **Ours** | | 4.70M | 66.5 | 64.5 | **72.3** | 57.0 | 70.5 | **70.0** | 70.6 | 68.0 | **75.1** |
| S:Source / T:Target | | | vKITTI/sKITTI | | | A2D2/sKITTI | | | A2D2/nuScenes | | |
| Strategy | Visual Backbone | Params | 2D | 3D | xM | 2D | 3D | xM | 2D | 3D | xM |
| Finetune | | 86.9M | 54.9 | 41.5 | 55.8 | 43.0 | 43.8 | 51.5 | 55.4 | 50.1 | 60.2 |
| Frozen | CLIP:ViT-B | 0.0M | 49.1 | 42.0 | 54.4 | 35.3 | 43.8 | 48.7 | 51.2 | 49.4 | 58.1 |
| **Ours** | | 1.82M | 55.6 | 43.6 | **58.0** | 43.2 | 44.6 | **52.0** | 56.3 | 50.3 | **61.0** |
| Finetune | | 305M | 57.4 | 43.5 | 58.7 | 46.9 | 44.3 | 53.0 | 57.2 | 50.8 | 61.3 |
| Frozen | CLIP:ViT-L | 0.0M | 54.0 | 42.9 | 58.4 | 41.8 | 44.4 | 51.4 | 53.7 | 50.0 | 59.6 |
| **Ours** | | 4.70M | 57.6 | 44.7 | **60.0** | 48.8 | 46.3 | **54.4** | 58.0 | 50.7 | **61.9** |

Table 3: Ablation study on the effectiveness of significant components in UniDSeg with the ViT-B backbone for DG3SS task.

| MTP | LST | nuScenes:Sing./USA | | | A2D2/sKITTI | | |
|---|---|---|---|---|---|---|---|
| | | 2D | 3D | xM | 2D | 3D | xM |
| Frozen VFM | | 58.3 | 67.9 | 71.2 | 35.3 | 43.8 | 48.7 |
| ✓ | | 63.9 | 67.8 | 72.5 | 40.4 | 44.0 | 50.5 |
| | ✓ | 65.7 | 67.8 | 73.3 | 41.8 | 44.2 | 51.1 |
| ✓ | ✓ | 68.2 | 68.0 | 74.0 | 43.2 | 44.6 | 52.0 |

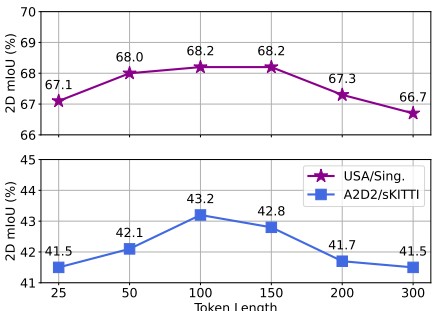

Figure 4: Effect of the learnable token length.

that due to the fixed and relatively small number of trainable parameters in the decode head, the count of trainable parameters presented in the tables is focused solely on the visual architecture. In this setup, we separately select CLIP:ViT-B and CLIP:ViT-L backbone with fine-tuning, freezing, and our proposed Learnable-parameter-inspired Mechanism for analysis. The results indicate that frozen VFMs outperform previous DA3SS methods without specialized design. In addition to "nuScenes:Day/Night", VFMs with full parameter fine-tuning exhibit enhanced performance relative to their frozen counterparts because the 3D modality is less sensitive to light, thus dominating in ensemble prediction direction. Remarkably, our method achieves even superior generalization capabilities, surpassing the full parameter fine-tuning 0.5∼2.7% mIoU with merely ∼2% trainable parameters compared to the original backbone. This ablation experiment demonstrates that our method can inherit the pre-existing target awareness from the VFMs to the maximum extent.

**Evaluation of Different Components.** As shown in Tab. 3, we train four models on two scenarios for DG3SS, including (1) Frozen parameter of ViT backbone; (2) only performing MTP in the frozen ViT by using prompt tuning, leading to a significant mIoU boost (1.3% and 1.8%). This highlights the robustness of the 3D-to-2D transitional prompts in cross-domain learning; (3) only performing LST in the frozen ViT by customizing a learnable token, demonstrating that learnable tokens can encourage the model to learn domain-invariant representations (increasing 2.1% and 2.4% mIoU); (4) combining two proposed components to reach peak value.

**Evaluation of Learnable Token Length.** We illustrate the impact of learnable token length $K$ on the overall 2D performance of UniDSeg with the ViT-B backbone for DG3SS. As shown in Fig. 4, the results demonstrate a consistent upward trend. When $K$ is set to 100, the relatively small

Table 4: Effect of using Segment Anything Model (SAM) as the 2D backbone.

| Task | 2D Backbone | USA/Sing. | | |
|------|-------------|-----------|------|------|
| | | 2D | 3D | xM |
| DG | CLIP:ViT-L | 66.5 | 64.5 | 72.3 |
| | SAM:ViT-L | 66.8 | 64.7 | 72.6 |
| DA | CLIP:ViT-L | 67.2 | 67.6 | 72.9 |
| | SAM:ViT-L | 67.8 | 68.8 | 73.3 |

Table 5: Effect of applying different training strategies to the SAM-based model.

| Task | Strategy | USA/Sing. | | |
|------|----------|-----------|------|------|
| | | 2D | 3D | xM |
| DG | Finetune | 65.9 | 64.3 | 70.8 |
| | Ours | 66.8 | 64.7 | 72.6 |
| DA | Finetune | 66.5 | 67.9 | 71.4 |
| | Ours | 67.8 | 68.8 | 73.3 |

Table 6: Effect of using different 3D backbones on the DA3SS methods.

| 3D Backbone | DA3SS | USA/Sing. | | |
|-------------|-------|-----------|------|------|
| | | 2D | 3D | xM |
| SparseConvNet | xMUDA | 64.4 | 63.2 | 69.4 |
| | UniDSeg | 67.2 | 67.6 | 72.9 |
| MinkowskiNet | xMUDA | 65.9 | 64.0 | 69.7 |
| | UniDSeg | 67.5 | 68.6 | 73.1 |

training parameters and high performance make it the preferred choice for subsequent experiments. Meanwhile, this observation suggests that the model benefits from incorporating visual information from multiple layers, enabling it to capture more nuanced and discriminative features.

**Evaluation of Different 2D and 3D Backbones.** The motivation of this work is to study a universal framework based on VFMs to enhance the generalizability and adaptability of cross-domain 3D semantic segmentation, demonstrating the effectiveness of the visual foundation model priors. However, CLIP is not designed for common surveillance-relevant tasks like semantic segmentation. Hereby, we dedicate ourselves to verifying its effectiveness on VFMs designed for segmentation tasks such as SAM [23]. Firstly, in Tab. 4, we evaluate the effect of SAM as the 2D backbone. It is observed that SAM-based UniDSeg exhibits better performance on "USA/Sing" scenario, with a "xM" gain of 0.3% on DG and 0.4% on DA. Then, in Tab. 5, we evaluate the effect of applying different training strategies to the SAM-based model. It is observed that our Learnable-parameter-inspired Mechanism for tuning the SAM-based encoder can achieve 1.8% and 1.9% "xM" gain compared to fine-tuning strategy. In addition, we evaluate UniDSeg on another 3D backbone, *i.e.,* MinkowskiNet, making it more convincing as a "universal" framework for cross-domain 3D semantic segmentation.

Table 7: The parameters and computational costs of CLIP-based and SAM-based 2D backbones. "Cost" means the percentage of trainable parameters in MTP and LST compared to fine-tuning the whole encoder consumed.

| 2D Backbone | Full Params | Trainable Params | Cost | MTP | LST |
|-------------|-------------|------------------|------|------|------|
| CLIP:ViT-B | 86.9M | 1.82M | **2.09%** | 0.48M | 1.34M |
| CLIP:ViT-L | 305M | 4.70M | **1.54%** | 1.78M | 2.92M |
| SAM:ViT-L | 307M | 4.34M | **1.41%** | 1.42M | 2.92M |

**Evaluation of Computation Cost.** In Tab. 7, we have reported model parameters of CLIP: ViT-B, CLIP: ViT-L, and SAM: ViT-L for the VFM-based encoder along with all trainable parameters. Note that, the entire ViT backbone in our VFM-based encoder is frozen during downstream training for DA3SS and DG3SS. Only two layer-wise learnable blocks, MTP and LST, are trainable. It is obvious that our method requires only 1-2% parameters optimization to exceed the fine-tuning results.

## 5    Conclusion

In this work, we delve into a universal method that can enhance the adaptability and generalizability of cross-domain 3D semantic segmentation, dubbed UniDSeg. For this purpose, we introduce a learnable-parameter-inspired mechanism to the off-the-shelf VFMs with frozen parameters, which maximally preserves pre-existing target awareness in VFMs, further enhancing the generalizability of VFMs. Our method achieves state-of-the-art performance in both DA3SS and DG3SS, as demonstrated by extensive experiments on different scenarios. We believe that our work will inspire a deeper investigation of cross-domain 3D semantic segmentation in autonomous driving.

**Acknowledgments.** This work was supported by the National Natural Science Foundation of China under Grant No.62176224, No.62306165; Natural Science Foundation of Chongqing under No.CSTB2023NSCQ-JQX0007; China Computer Federation (CCF) Lenovo Blue Ocean Research Fund; China Academy of Railway Sciences No.2023YJ357.

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

# A   Appendix / supplemental material

This section starts with more information on UniDSeg, including dataset split, extensible tasks, more ablation studies, and additional visualization results.

## A.1   Dataset Split of Cross-domain Learning

To compose our domain adaptive and generalized scenarios, following [19], we exploit public datasets, including nuScenes [4], VirtualKITTI [11], SemanticKITTI [3], and A2D2 [12]. The split details are tabulated in Tab. 8.

Table 8: Size of the splits in frames for all proposed cross-domain learning scenarios.

| Scenarios | Source Train | Target Train | Target Val/Test | Categories |
|---|---|---|---|---|
| nuScenes:Day/Night | 24745 | 2779 | 606/602 | **Vehicle**: [bicycle, bus, car, construction_vehicle, motorcycle, trailer, truck]; |
| nuScenes:USA/Sing. | 15695 | 9665 | 2770/2929 | **Driveable Surface**; **Sidewalk**; **Terrain**; **Manmade**; **Vegetation** |
| nuScenes:Sing./USA | 9665 | 15695 | -/3090 | |
| vKITTI/sKITTI | 2126 | 18029 | 1101/4071 | vKITTI: **Car**; **Truck**; **Road**; **Object**: [traffic sign, traffic light, pole, misc]; **Building**; **Vegetation**: [terrain, tree, vegetation] 
 sKITTI: **Car**; **Truck**; **Road**; **Object**: [fence, pole, traffic-sign, other-object]; **Building**; **Vegetation**: [vegetation, trunk, terrain] |
| A2D2/sKITTI | 27695 | 18029 | 1101/4071 | A2D2: **Car**; **Truck**; **Bike**: [bicycle, small vehicle]; **Person**; **Road**; **Parking**; **Sidewalk**: [sidewalk, curbstone]; **Object**; **Building**; **Vegetation** 
 sKITTI: **Car**; **Truck**; **Bike**: [bicycle, motorcycle, bicyclist, motorcyclist]; **Person**; **Road**; **Parking**; **Sidewalk**; **Object**; **Building**; **Vegetation**: [terrain, trunk, vegetation] |
| A2D2/nuScenes | 27695 | 25330 | 2800/6019 | A2D2: **Car**; **Truck**; **Bike**: [bicycle, small vehicle]; **Person**; **Road**; **Sidewalk**: [sidewalk, curbstone]; **Building**; **Vegetation** 
 nuScenes: **Car**; **Truck**; **Bike**: [bicycle, motorcycle]; **Person**; **Road**: [driveable_surface]; **Sidewalk**; **Building**: [manmade]; **Vegetation**: [terrain, vegetation] |

**nuScenes.**   It contains 1,000 scenes, each of 20 seconds, corresponding to 40k annotated keyframes taken at 2Hz. The original scenes are split into 28,130 training frames and 6,019 validation frames. Each frame contains a 32-beam LiDAR point cloud with point-wise annotations and six RGB images captured by six cameras from different views of LiDAR. For nuScenes:Day/Night, we choose 602 night scenes for testing data, while for nuScenes:USA/Sing. and nuScenes:Sing./USA, we choose 2,929 Singapore and 3,090 USA scenes for testing data, respectively. Both of them merge the objects into 6 categories: **Vehicle**, **Driveable Surface**, **Sidewalk**, **Terrain**, **Manmade**, and **Vegetation**.

**VirtualKITTI.**   It consists of 5 driving scenes which are created with the Unity game engine by real-to-virtual cloning of the scenes 1, 2, 6, 18, and 20 of the real KITTI dataset. Different from real KITTI, VirtualKITTI does not simulate LiDAR, but rather provides a dense depth map, alongside semantic, instance, and flow ground truth. Each of the 5 scenes contains between 233 and 837 frames, *i.e.*, in total 2126 for the 5 scenes. Each frame is rendered with 6 different weather/lighting variants (clone, morning, sunset, overcast, fog, rain) which we use all.

**SemanticKITTI.**   It is a large-scale dataset based on the KITTI Odometry Benchmark captured in Germany. The original scenes are split into 19,130 training scans and 4,071 validation scans. Unlike nuScenes, SemanticKITTI only provides the front-view images and a 64-layer front LiDAR. 19 categories are used for segmentation.

**A2D2.**   It consists of 20 drives, which corresponds to 28,637 frames. The point cloud comes from three 16-layer front LiDARs (center, left, and right), where the left and right LiDARs are inclined. By projecting 3D point clouds onto 2D images, corresponding 2D semantic labels are regarded as 3D point-wise labels, which contain 38 categories.

Note that, we select 6 merged categories between the VirtualKITTI and SemanticKITTI, including **Car**, **Truck**, **Road**, **Object**, **Building**, and **Vegetation**. Between A2D2 and nuScenes, we select 8 merged categories, including **Car**, **Truck**, **Bike**, **Person**, **Road**, **Sidewalk**, **Building**, and **Vegetation**. Between A2D2 and SemanticKITTI, 2 additional merged categories are considered, which are **Object**, and **Parking**.

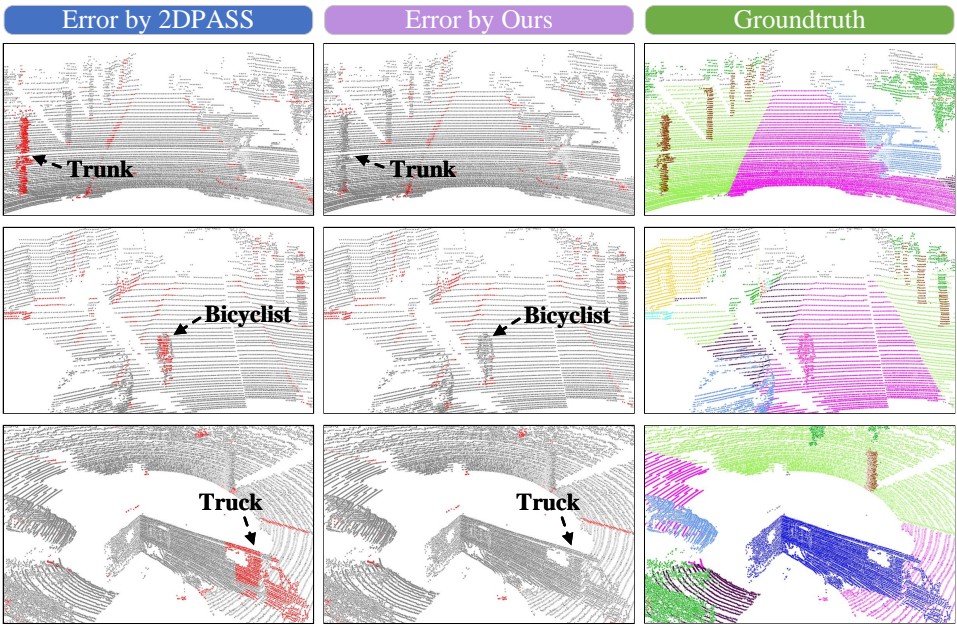

Figure 5: Qualitative results of our method on the validation set of SemanticKITTI. The misclassification points are signed in red.

## A.2 Fully-supervised 3D Semantic Segmentation

The proposed method can also work reliably in handling fully-supervised large-scale 3D semantic segmentation. As shown in Tab. 9, we simply compare the results of some typical uni-modal and multi-modal methods, including MinkowskiNet [7], SPVCNN [42], Cylinder3D [62], and 2DPASS [52], on the publicly available *SemanticKITTI* validation set. According to the official setting in 00-10 sequences, sequence 08 is the validation set, while the remaining sequences are the training set.

In this experimental setup, we make some slight modifications to the training strategy. Considering that fully-supervised learning assumes the training and validation/testing sets are independent identically distribution, without involving domain shift problems in cross-domain learning. (*i.e.*, avoid overfitting the source-domain data distribution). Therefore, when training our model, we replace the 2D backbone in 2DPASS with our VFMs backbone and fine-tune its parameters in the same way. Moreover, we reduce the learning rate of the 2D backbone to $lr = 0.024$ while keeping the learning rates of the other modules $lr = 0.24$ and the number of training epochs $ep = 64$ unchanged to ensure the fairness of the experiment. Similar to 2DPASS, we tabulate the validation results with and without Test-Time Augmentation (TTA) and set the number of views to 12 as the default. Some visualization results are shown in Fig. 5. 2DPASS has a higher error recognizing small objects and region boundaries, while our method recognizes small objects better thanks to the knowledge prior of visual foundation models.

## A.3 Source-Free Domain Adaptive 3D Semantic Segmentation

Source-free domain adaptation (SFDA) seeks to learn models where the *vendor* can trade only the source model and the *client* can perform target adaptation without accessing source-domain data. Drawing inspiration from Jogendra *et al.* [25], SFDA enables partition into two tasks: (a) source-only domain generalization and (b) source-free target adaptation. Hereby, towards the former, we utilize UniDSeg to achieve models with generalization capability. Towards the latter, we utilize target 3D pseudo-labels (PL) obtained from the source model for self-training. When selecting PL from the target-domain data, we consider the ensemble result "xM" as the fusion PL to supervise both the 2D and 3D branches. As shown in Tab. 10, we adopt conventional consistency learning and pseudo-label self-training to address the source-free domain adaptive 3D semantic segmentation

Table 9: Fully-supervised 3D semantic segmentation results on the *SemanticKITTI* validation set. We report per-class IoU. "†" denotes the reproduced result referring to the official codebase. "w/ TTA" means using test-time augmentation in the inference stage.

| Method | car | bicycle | motorcycle | truck | bus | person | bicyclist | motorcyclist | road | parking | sidewalk | other-ground | building | fence | vegetation | trunk | terrain | pole | traffic-sign | mIoU (%) |
|---|---|---|---|---|---|---|---|---|---|---|---|---|---|---|---|---|---|---|---|---|
| MinkowskiNet [7] | - | - | - | - | - | - | - | - | - | - | - | - | - | - | - | - | - | - | - | 61.1 |
| SPVCNN [42] | - | - | - | - | - | - | - | - | - | - | - | - | - | - | - | - | - | - | - | 63.8 |
| Cylinder3D [62] | - | - | - | - | - | - | - | - | - | - | - | - | - | - | - | - | - | - | - | 65.9 |
| 2DPASS† [52] | 95.3 | 47.1 | 73.7 | 81.8 | 56.0 | 73.5 | 87.6 | 2.1 | 92.4 | 45.2 | 78.6 | 1.0 | 90.8 | 61.8 | 88.4 | 69.5 | 75.5 | 58.1 | 51.8 | 64.7 |
| 2DPASS† [52] w/ TTA | 96.6 | 52.2 | 77.9 | 91.1 | 68.2 | 77.9 | 92.0 | 0.2 | 94.0 | 50.6 | 81.4 | 1.2 | 91.8 | 66.3 | 89.6 | 72.0 | 77.3 | 63.0 | 53.5 | 68.2 |
| **Ours** | 96.2 | 47.2 | 70.1 | 84.3 | 64.5 | 74.1 | 89.5 | 2.1 | 92.6 | 46.6 | 79.1 | 3.2 | 90.9 | 62.8 | 88.3 | 69.9 | 75.1 | 58.6 | 52.1 | **65.6**$_{+0.9}$ |
| **Ours** w/ TTA | 97.0 | 52.3 | 73.4 | 92.6 | 71.1 | 78.3 | 92.3 | 0.0 | 94.1 | 51.3 | 81.8 | 3.3 | 92.1 | 67.4 | 89.5 | 72.0 | 77.0 | 63.8 | 54.6 | **68.6**$_{+0.4}$ |

Table 10: Performance comparison of SFDA3SS methods in three typical scenarios. "†" denotes the reproduced result referring to the official codebase, as the different category splits applied in the same adaptation scenario.

| | S:Source / T:Target | | nuScenes:USA/Sing. | | | nuScenes:Day/Night | | | A2D2/sKITTI | | |
|---|---|---|---|---|---|---|---|---|---|---|---|
| Task | Method | Source-free | 2D | 3D | xM | 2D | 3D | xM | 2D | 3D | xM |
| DA | Baseline | ✓ | 58.4 | 62.8 | 68.2 | 47.8 | 68.8 | 63.3 | 34.2 | 35.9 | 40.4 |
| | Consistency | ✓ | 58.7 | 63.2 | 68.1 | 50.4 | 66.8 | 63.6 | 37.1 | 36.5 | 41.8 |
| | Pseudo-Label | ✓ | 58.9 | 62.7 | 68.5 | 48.3 | 69.0 | 63.2 | 37.6 | 36.6 | 41.5 |
| | SUMMIT† [41] | ✓ | 61.6 | 66.2 | 68.4 | 53.8 | 68.9 | 68.2 | 42.9 | 43.7 | 46.8 |
| | **UniDSeg** | ✗ | 67.2 | 67.6 | 72.9 | 63.2 | 71.2 | 71.2 | 50.7 | 55.4 | 57.5 |
| | **UniDSeg** | ✓ | 69.3 | 71.7 | 73.5 | 62.6 | 70.7 | 68.7 | 49.6 | 59.1 | 58.6 |

(SFDA3SS). Moreover, we compare UniDSeg under the source-free condition with the pioneering method SUMMIT [41]. Experimental results demonstrate that our method achieves superior performance, remarkably on "USA/Sing." and "A2D2/sKITTI". Note that we do not adopt any source-free adaptive learning methods, but only extend the DG3SS model to SFDA3SS tasks, to prove that a good generalization model is conducive to the execution of SFDA3SS. We have reason to believe that well-designed source-free adaptive learning can bring even more performance improvements.

### A.4   Re-training DA3SS with Pseudo-Label

In general, cross-modal learning and self-training with pseudo-labels are complementary in their combination. As shown in Tab. 11, when re-train the DA3SS model with PL, our method still achieves competitive performance. Similar to SFDA3SS, when selecting PL from the target-domain data, we consider the ensemble result "xM" as the fusion PL to supervise both the 2D and 3D branches.

**Limitation.**   Nevertheless, compared to other methods that achieve performance improvements of 1∼2% mIoU in 2D prediction with the help of target PL, our method only improves via the 3D pseudo supervision of the target domain, with almost no impact on the 2D prediction. We speculate that the proposed learnable-parameter-inspired mechanism is more effective in unleashing the potential of VFMs, providing more pre-existing target information. With the increase in reliable target label information, the limited learnable parameters cannot bear more contextual information, which hinders the "1+1=2" effect. This is the problem that our unified cross-domain 3D semantic segmentation model needs to be improved in the future.

### A.5   Evaluation on the Roles of Sparse Depth

MM2D3D [6] provides an effective approach to solving the DA3SS task by utilizing depth as an auxiliary input. By adding a parallel reinitialized encoder to the 2D backbone to process the sparse depth obtained from the point cloud. Therefore, we consider analyzing whether the sparse depth needs to be deeply encoded or employed as a prompt. As shown in Fig. 6, we illustrate the architecture of two learning manners: (a) *Depth Deep Encoding* and (b) *Depth as Point-level Prompts*.

Table 11: Performance comparison of multi-modal domain adaptive 3D semantic segmentation methods with pseudo-label ("$_{PL}$") re-training on four typical scenarios.

| S:Source / T:Target | | nuScenes:USA/Sing. | | | nuScenes:Day/Night | | | vKITTI/sKITTI | | | A2D2/sKITTI | | |
|---|---|---|---|---|---|---|---|---|---|---|---|---|---|
| Task | Method | 2D | 3D | xM | 2D | 3D | xM | 2D | 3D | xM | 2D | 3D | xM |
| DA | xMUDA$_{PL}$ [19] | 67.0 | 65.4 | 71.2 | 57.6 | 69.6 | 64.4 | 45.8 | 51.4 | 52.0 | 41.2 | 49.8 | 47.5 |
| | AUDA$_{PL}$ [30] | 65.9 | 65.3 | 70.6 | 54.3 | 69.6 | 61.1 | 35.9 | 45.5 | 45.9 | 46.8 | 48.1 | 50.6 |
| | DsCML$_{PL}$ [35] | 65.6 | 57.5 | 66.9 | 51.4 | 49.8 | 53.8 | 39.6 | 41.8 | 42.2 | 46.8 | 51.8 | 52.4 |
| | Dual-Cross$_{PL}$ [28] | 66.5 | 59.8 | 68.8 | 59.1 | 69.8 | 68.2 | 43.1 | 39.4 | 47.6 | 44.9 | 52.8 | 52.3 |
| | SSE$_{PL}$ [58] | 66.9 | 64.4 | 70.6 | 59.1 | 67.0 | 66.3 | 47.2 | 53.5 | 55.2 | 45.9 | 51.5 | 52.5 |
| | BFtD$_{PL}$ [45] | 65.9 | 66.0 | 71.3 | 60.6 | 70.0 | 66.6 | 48.6 | 55.4 | 57.5 | 42.6 | 53.7 | 52.7 |
| | MM2D3D$_{PL}$ [6] | 74.3 | 68.3 | **74.9** | 71.3 | 69.6 | **72.2** | 55.4 | 55.0 | 59.7 | 46.4 | 48.7 | 50.7 |
| | **UniDSeg**$_{PL}$ | 67.4 | 70.0 | 73.0 | 64.5 | 71.9 | 71.7 | 60.5 | 52.7 | **62.7** | 50.4 | 57.3 | **58.6** |

For (a), similar to MM2D3D, we replace the 2D backbones with two ViT-B models. The sparse depth is fed into the ViT-B that trains from scratch, while the image is fed into the ViT-B pre-trained on large-scale datasets. After that, we can obtain encoded output by concatenating two representations. For (b), that is the simple architecture of only sparse depth as prompt for fine-tuning. Tab. 12 shows the comparison results of the aforementioned learning manners. Compared to (a), (b) not only requires 0.6% fine-tuning parameters of the 2D backbone but also achieves superior segmentation performance, making it flexible to expand to various 3D semantic segmentation tasks with multi-modal learning.

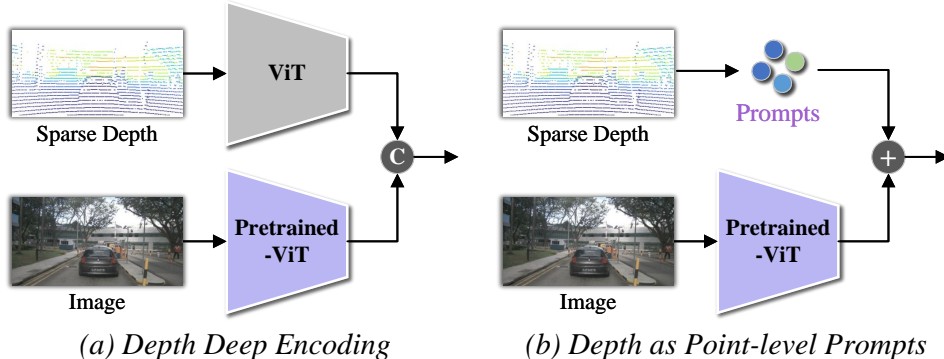

(a) Depth Deep Encoding    (b) Depth as Point-level Prompts

Figure 6: Role of Depth with brief diagram.

Table 12: Performance comparison of two sparse depth learning roles for DG3SS.

| Role of Depth | Params | nuScenes:Sing./USA | | |
|---|---|---|---|---|
| | | 2D | 3D | xM |
| Deep Encoding | 86.9M | 66.1 | 67.7 | 73.0 |
| Point-level Prompts | 0.48M | 67.8 | 67.9 | 73.8 |

## A.6    Additional Visualization Results

In this part, we demonstrate more qualitative results of our proposed UniDSeg to illustrate the effectiveness of our DA3SS framework. The corresponding qualitative results from 2D and 3D predictions will also be provided (See Figs 7, 8, 9, and 10). Note that the prediction differences are signed in the red rectangle.

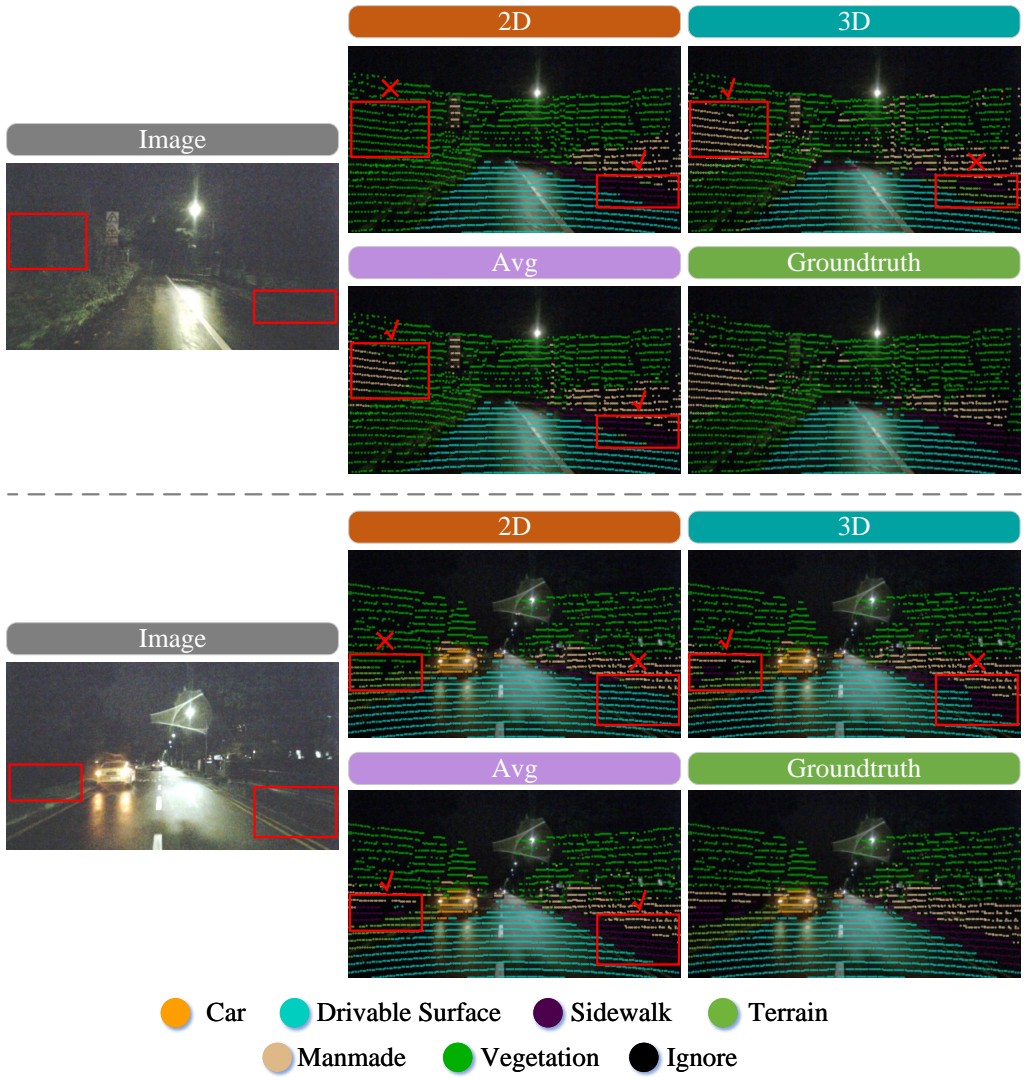

Figure 7: Additional qualitative results of *nuScenes:Day/Night* scenario for DA3SS.

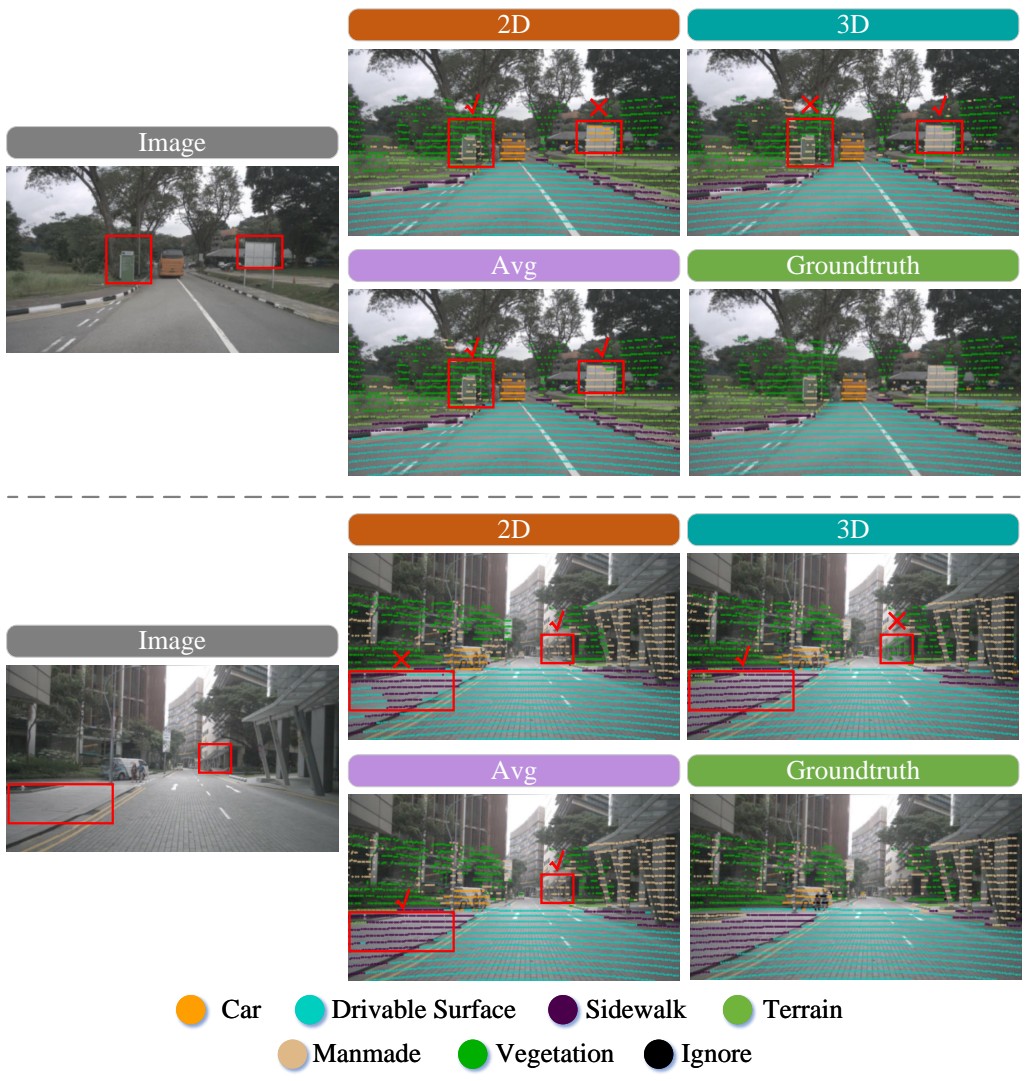

Figure 8: Additional qualitative results of *nuScenes:USA/Sing.* scenario for DA3SS.

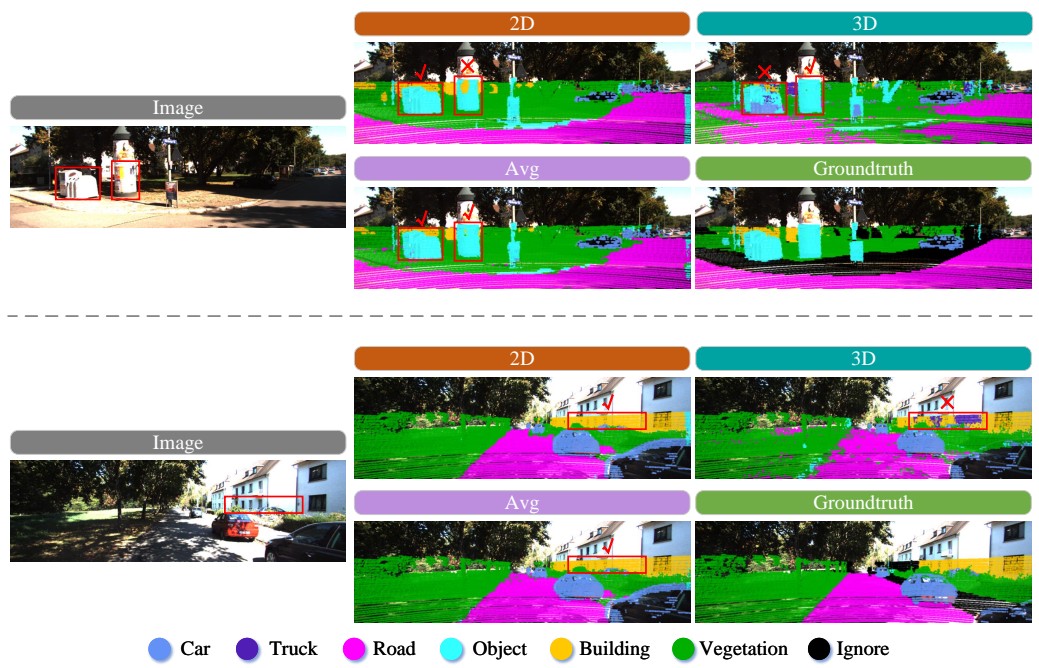

Figure 9: Additional qualitative results of *vKITTI/sKITTI* scenario for DA3SS.

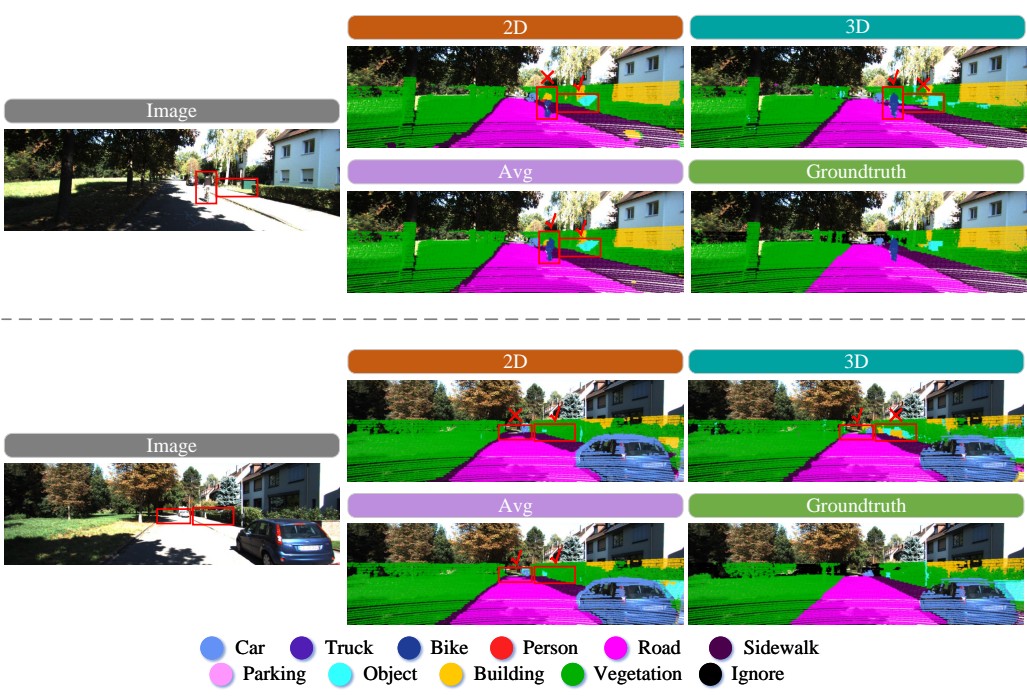

Figure 10: Additional qualitative results of *A2D2/sKITTI* scenario for DA3SS.

