# OpenReview forum: "UniDSeg: Unified Cross-Domain 3D Semantic Segmentation via Visual Foundation Models Prior"
_NeurIPS.cc/2024/Conference — NeurIPS 2024 poster_

### Official Review · Reviewer_T5Kw · 2024-07-03

**Soundness:** 3
**Presentation:** 3
**Contribution:** 2
**Rating:** 4
**Confidence:** 4

**Summary:**

This paper presents UniDSeg, a universal approach that enhances the adaptability and generalizability of cross-domain 3D semantic segmentation. To achieve this, we propose a learnable parameter-inspired mechanism for off-the-shelf VFMs with frozen parameters. This mechanism maximally preserves the pre-existing target awareness in VFMs, thereby further enhancing their generalizability.

**Strengths:**

1.	The motivation is clear.
2.	The proposed method is intuitive, and the experiments have validated their contributions.

**Weaknesses:**

1.	Since the backbone is modified from xMUDA, the oracle performance should also be provided.
2.	The task settings in this study are derived from xMUDA (TPAMI), while the prompt tuning and corresponding experimental settings are based on VPT. Therefore, this proposed method can be considered somewhat incremental.

**Questions:**

1.	Please explain why token length is equal to 100, the best performance can be obtained. However, it is important to note that the prompt tuning process may not be entirely stable. To address this, all experiments should be conducted multiple times, and the average performance along with the standard deviation also should be computed and reported.
2.	Please explain the main contribution compared with “Learning to Adapt SAM for Segmenting Cross-domain Point Clouds”.
3.	Since VFM and ViT are utilized as encoder, the model parameters and computational costs should be comprehensive reported.
4.	Did the authors notice that when the point labels projected onto 2D images, there are slight mismatch between these labels and 2D image pixels? Please discuss the potential reasons.

**Limitations:**

1.	The authors have addressed the limitations and potential negative societal impact of their work.

---

> ### Author Rebuttal · Authors · 2024-08-07
>
> Dear Reviewer T5Kw,
>
> We thank you for the precious review time and valuable comments. We have provided corresponding responses and results, which we believe have covered your concerns.
>
> We hope to further discuss with you whether your concerns have been addressed or not. If you still have any unclear parts of our work, please let us know. Thanks.
>
> Best,
> Authors
>
> **Q1**: Since the backbone is modified from xMUDA, the oracle performance should also be provided.
>
> **A1**: "Oracle" means solely training on the target domain, except in the "Day/Night" case where it uses a 50%/50% source/target batches to prevent overfitting due to the small target size, thereby serving as the upper bound for DA3SS.
> | | USA/Sing. | | | Day/Night | | | vKITTI/sKITTI | | | A2D2/sKITTI | | |
> |-|:-:|:-:|:-:|:-:|:-:|:-:|:-:|:-:|:-:|:-:|:-:|:-:|
> | | 2D | 3D | Avg | 2D | 3D | Avg | 2D | 3D | Avg | 2D | 3D | Avg |
> | Source-only | 58.4 | 62.8 | 68.2 | 47.8 | 68.8 | 63.3 | 26.8 | 42.0 | 42.2 | 34.2 | 35.9 | 40.4 |
> | UniDSeg | 67.2 | 67.6 | 72.9 | 63.2 | 71.2 | 71.2 | 60.5 | 50.9 | 62.0 | 50.7 | 55.4 | 57.5 |
> | Orcale| 75.4 | 76.0 | 79.6 | 61.5 | 69.8 | 69.2 | 66.3 | 78.4 | 80.1 | 59.3 | 71.9 | 73.6 |
>
> **Q2**: The task settings in this study are derived from xMUDA (TPAMI), while the prompt tuning and corresponding experimental settings are based on VPT. Therefore, this proposed method can be considered somewhat incremental.
>
> **A2**: Our method is groundbreaking in introducing the prompt-tuning concept into the **universal model for DG3SS and DA3SS**. In VPT-Deep, each prompt is a learnable token randomly initialized and directly insert into input space of all ViT layers. In contrast, **our method first leverages sparse depth as a point-level prompts inserted to each ViT layers**. It not only learns spatial distance perception prompts from point clouds but also learns invariance to sample perturbations. Secondly, **our method uses the learnable token as query for seeking matched prompting** after encoding in the ViT layer. It bridges the discrepancy between the information of pre-training dataset and the target scene.
>
> **Q3**: Please explain why token length is equal to 100, the best performance can be obtained. The average performance along with the standard deviation also should be computed and reported.
>
> **A3**: Consistent with all competitors, for each run of the experiment, we select the best model based on the validation set results (Via trial and error, when token length is set to 100, the result is stable and up to peak value), and use it for the final inference on the test set. All experimental settings are conducted at least 3 times and we select the best performance of all runs. The table below reports the 2D performance with standard deviation over 4 runs when the token length is set to 100.
> | | USA/Sing. | A2D2/sKITTI |
> |-|:-:|:-:|
> | 2D mIoU | 68.100&plusmn;0.071 | 43.975&plusmn;0.453 |
>
> **Q4**: Please explain the main contribution compared with "Learning to Adapt SAM for Segmenting Cross-domain Point Clouds".
>
> **A4**: Above work leverages whole SAM to generate instance mask, guiding the alignment of features from diverse 3D data domains into a unified domain. On the contrary, our proposed UniDSeg does not directly use instance masks generated offline, but instead uses only the pre-trained image encoder of VFMs (e.g., CLIP and SAM) to extract visual prior information. UniDSeg is groundbreaking in introducing the prompt-tuning concept into the universal model for DG3SS and DA3SS. We propose a learnable-parameter-inspired mechanism to the off-the-shelf image encoder of VFMs, which maximally preserves pre-existing target awareness in VFMs to further enhance its generalizability. Just as the following A5 response, our method not only requires nearly 2% fine-tuning parameters of the visual backbone but also achieves superior segmentation performance.
>
> **Q5**: Since VFM and ViT are utilized as encoder, the model parameters and computational costs should be comprehensive reported.
>
> **A5**: Thank you for your suggestion. We have reported model parameters of CLIP: ViT-B, CLIP: ViT-L, and SAM: ViT-L for the VFM based encoder along with all trainable parameters. Note that, the entire ViT backbone in our VFM-based encoder is frozen during downstream training for DA3SS and DG3SS. Only two layer-wise learnable blocks, MTP and LST, are trainable. Hereby, we provide supplementary information on the trainable parameters for MTP and LST in the following table, where "Cost" means only need percentage trainable parameters compared to fine-tuning whole encoder consume.
> | Visual Backbone | VFM-based Encoder | All trainable Params | **Cost** | MTP | LST |
> |-|:-:|:-:|:-:|:-:|:-:|
> | CLIP: ViT-B | 86.9M | 1.82M | **2.09%** | 0.48M | 1.34M |
> | CLIP: ViT-L | 305M | 4.70M | **1.54%** | 1.78M | 2.92M |
> | SAM: ViT-L | 307M | 4.34M | **1.41%** | 1.42M | 2.92M |
>
> **Q6**: Did the authors notice that when the point labels projected onto 2D images, there are slight mismatch between these labels and 2D image pixels? Please discuss the potential reasons.
>
> **A6**: The mismatch can be caused by a variety of factors, including calibration error, geometric error, image resolution, etc. These factors are beyond the scope of this work. Following xMUDA, we assume that the calibration of LiDAR-Camera is available for both domains and does not change over time. Since images and point cloud are heterogeneous, according to extrinsic matrix from dataset, only the points falling into the intersected field of view are geometrically associated with multi-modal data (i.e., 3D-to-2D projection).

---

> > ### Comment · Reviewer_T5Kw · 2024-08-10
> > **Reviewer response**
> >
> > I've reviewed the authors' responses and appreciate their engagement. I will stay in touch for further discussion as we approach the final rating.

---

> > > ### Comment · Reviewer_T5Kw · 2024-08-12
> > > **Official Comment by Reviewer T5Kw**
> > >
> > > Reviewer 4rFB mentions a new question, please carefully answer it since the discussion is about to end.

---

> > > > ### Author Response · Authors · 2024-08-12
> > > >
> > > > Thank you for your reminder. We have answered the question of Reviewer 4rFB.

---

### Official Review · Reviewer_4rFB · 2024-07-12

**Soundness:** 4
**Presentation:** 2
**Contribution:** 2
**Rating:** 5
**Confidence:** 4

**Summary:**

This paper introduces the prompt-tuning concept into DG3SS and DA3SS, and proposes a learnable parameter heuristic mechanism for the off-the-shelf VFM. Modal Transitional Prompting is proposed to capture 3D-to-2D transitional prior and task-shared knowledge from the prompt space. Learnable Spatial Tunability is constructed to the representation of distinct instances driven by prompts in the query space. Extensive experimental results demonstrate the effectiveness of the proposed method on widely recognized tasks and datasets.

**Strengths:**

1. The proposed method of improving cross-domain 3D semantic segmentation based on visual basic models is interesting and innovative.
2. The proposed Modal Transitional Prompting (MTP) and Learnable Spatial Tunability (LST) explore and utilize 2D prior knowledge from VFMs.
3. Experiments on multiple datasets verify the effectiveness of the method.

**Weaknesses:**

1. The method lacks theoretical analysis. In Section 3.2, Modal Transitional Prompting, line 193, the author mentioned that "PG is designed to capture 3D-to-2D transitional prior and task-shared knowledge from the prompt space." However, the designed method only uses learnable parameters, which is not related to the 3D-to-2D transitional prior and the promotion of 3D domain adaptation. This section only explains the steps of the method and lacks theoretical explanation of the method.

2. In Section 3.2, Learnable Spatial Tunability, the author mentioned that it was inspired by Rein [1]. However, it seems that the proposed LST has a similar design to Rein, which casts doubt on the novelty of LST. The modification of Rein lacks innovation and theoretical analysis. This inevitably makes people think that the good performance of LST is due to the effectiveness of Rein-like components.

3. In the experimental section, VFM is only verified based on the classification model CLIP [2]. However, the CLIP paper [2] states that “Additionally, CLIP is not designed for common surveillance-relevant tasks like object detection and semantic segmentation. This means it has limited use for certain surveillance tasks”. The proposed method is dedicated to segmentation tasks, but it is only verified on the classification model CLIP. As a pioneer method, it is recommended to verify its effectiveness on VFMs designed for segmentation tasks such as SAM [3] and SEEM [4].

[1]Wei, Zhixiang, Lin Chen, Yi Jin, Xiaoxiao Ma, Tianle Liu, Pengyang Ling, Ben Wang, Huaian Chen, and Jinjin Zheng. "Stronger Fewer & Superior: Harnessing Vision Foundation Models for Domain Generalized Semantic Segmentation." In Proceedings of the IEEE/CVF Conference on Computer Vision and Pattern Recognition, pp. 28619-28630. 2024.

[2]Alec Radford, Jong Wook Kim, Chris Hallacy, Aditya Ramesh, Gabriel Goh, Sandhini Agarwal, Girish Sastry, Amanda Askell, Pamela Mishkin, Jack Clark, et al. Learning transferable visual models from natural language supervision. In ICML, pages 8748–8763, 2021.

[3]Alexander Kirillov, Eric Mintun, Nikhila Ravi, Hanzi Mao, Chloe Rolland, Laura Gustafson, Tete Xiao, Spencer Whitehead, Alexander C. Berg, Wan-Yen Lo, Piotr Dollar, and Ross Girshick. Segment anything. In Proceedings of the IEEE/CVF International Conference on Computer Vision (ICCV), pages 4015–4026, 2023. 1, 2, 3, 6, 11

[4]Zou, Xueyan, Jianwei Yang, Hao Zhang, Feng Li, Linjie Li, Jianfeng Wang, Lijuan Wang, Jianfeng Gao, and Yong Jae Lee. "Segment everything everywhere all at once." Advances in Neural Information Processing Systems 36 (2023).

**Questions:**

1. In Section 3.2, Modal Transitional Prompting, line 199, the author mentioned "From the view of deep encoding, it focuses on the scope of scenes at different scales, so that their corresponding features have different content representations when constructed." Why does sparse depth have scenes at different scales?
2. Why does the model require multiple layers of PG and TB? Experiments are needed to verify the impact of the number of layers on the results.
3. How long does it take from the start of training to completion?
4. In Table 1, caption introduction: "Avg" is obtained by averaging the predicted probabilities from the 2D and 3D networks. However, "Avg" is ambiguous here. This leads to misleading averages of 2D and 3D mIou results, but this may not be the case in reality. It is recommended to replace "Avg" with another representation.

**Limitations:**

As above

---

> ### Author Rebuttal · Authors · 2024-08-07
>
> Dear Reviewer 4rFB,
>
> We thank you for the precious review time and valuable comments. We have provided corresponding responses and results, which we believe have covered your concerns.
>
> We hope to further discuss with you whether your concerns have been addressed or not. If you still have any unclear parts of our work, please let us know. Thanks.
>
> Best,
> Authors
>
> **Q1**: This section only explains the steps of the method and lacks theoretical explanation of the method.
>
> **A1**: Firstly, sparse depth map as a perspective projection representation of point clouds presents an unnatural image that can be processed by the image encoder in VFM. This is an additional prior derived from the point cloud. Lee et al. [Ref1] have been proved that source data internally know a lot more about the world and how the scene is formed, which called Privileged Information. This Privileged Information includes physical properties (e.g., depth) that might be useful for cross-domain learning. Secondly, though depth information is easy-to-access and tightly coupled with semantic information, Hu et al. [Ref2] argue that depth clues that complement colors are hard to deduce from color images alone, thus directly deep encoding fail to capture valid geometric information. To this end, we consider utilizing depth as point-level prompts inserted into ViT layers, enabling the model to learn spatial distance perception. This complements 2D representations, as LiDAR-derived depth information is less affected by domain variations compared to images, which can be easily influenced by changes in lighting and other factors. Experimental results in Table 8 further demonstrate the effectiveness of our method. For convenience, we reshow the table below:
> | Role of Depth | Params | 2D | 3D | Ens|
> |-|:-:|:-:|:-:|:-:|
> | (a) Deep Encoding | 86.9M | 66.1 | 67.7 | 73.0 |
> | (b) Point-level Prompts | 0.48M | 67.8 | 67.9 | 73.8 |
>
> [Ref1] Lee K H, Ros G, Li J, et al. Spigan: Privileged adversarial learning from simulation. arXiv preprint arXiv:1810.03756, 2018.
>
> [Ref2] Hu S, Bonardi F, Bouchafa S, et al. Multi-modal unsupervised domain adaptation for semantic image segmentation. Pattern Recognition, 2023.
>
> **Q2**: The novelty of LST.
>
> **A2**: The innovation of our method is in introducing learnable-parameter-inspired mechanism to the off-the-shelf VFMs, which is guided by point-level prompts from 3D information. To achieve this, we place layer-wise MTP and LST blocks to take full advantage of semantic understanding of diverse levels and modalities. Among them, the affinity matrix computed in LST is a common solution to capture the associations between learnable tokens and 2D representations. The insight of learnable token originally comes from Visual Prompt Tuning (VPT) [Ref3].
>
> [Ref3] Jia M, Tang L, Chen B C, et al. Visual prompt tuning. European Conference on Computer Vision, 2022: 709-727.
>
> **Q3**: As a pioneer method, it is recommended to verify its effectiveness on VFMs designed for segmentation tasks such as SAM [3] and SEEM [4].
>
> **A3**: We have supplemented the experimental results for DA3SS and DG3SS under another visual backbone, SAM: ViT-L. Notably, in this work, we only utilize the off-the-shelf image encoder of VFMs (e.g., CLIP (w/o text encoder) and SAM (w/o prompt encoder and mask decoder)). As shown in the table below, SAM-based UniDSeg exhibit better performance on "USA/Sing." scenario. Due to rebuttal time limitation, we will supplement all SAM-based experimental results subsequently.
> | Task | Method | Visual Backbone | 2D | 3D | Ens |
> |-|:-:|:-:|:-:|:-:|:-:|
> | DG | UniDSeg | CLIP: ViT-L | 66.5 | 64.5 | 72.3 |
> | | | **SAM: ViT-L** | 66.8 | 64.7 | **72.6 (+0.3)** |
> | DA | UniDSeg | CLIP: ViT-L | 67.2 | 67.6 | 72.9 |
> | | | **SAM: ViT-L** | 67.8 | 68.8 | **73.3 (+0.4)** |
>
> **Q4**: Why does sparse depth have scenes at different scales?
>
> **A4**: We are sorry to make you misunderstand. Here, "different scales" refers to different receptive fields of adjacent pixels. The complementarity of the features extracted by the 2D and 3D branches is also tightly correlated to the different information processing machinery, i.e., 2D and 3D convolutions, which makes networks focusing on different areas of the scene with different receptive fields. When sparse depth is projected onto the image plane and processed via a 2D network, it can focus on the scope of scenes at different scales. For example, adjacent pixels in the image may have similar pixel values, but their corresponding depth values (distance) in the depth map can differ significantly.
>
> **Q5**: Experiments are needed to verify the impact of the number of layers of MTP and LST.
>
> **A5**: In the following table, "Half layer" indicates that the MTP and LST blocks are inserted every other layer. Due to rebuttal time limitation, we will supplement the experiments of other layer combinations subsequently.
> | Visual Backbone | MTP | LST | 2D | 3D | Ens|
> |-|:-:|:-:|:-:|:-:|:-:|
> | CLIP: ViT-L | All layers | All layers | 66.5 | 64.5 | 72.3 |
> | | Half layers | Half layers | 66.1 | 64.3 | 71.9 |
>
> **Q6**: How long does it take from the start of training to completion?
>
> **A6**: All experiments are conducted on one NVIDIA RTX 3090 GPU with 24GB RAM.
> | Task | USA/Sing. | Day/Night | vKITTI/sKITTI | A2D2/sKITTI |
> |-|:-:|:-:|:-:|:-:|
> | DA | ~22h | ~37h | ~21h | ~48h |
> | DG | ~12h | ~21h | ~11h | ~27h |
>
> **Q7**: "Avg" is ambiguous. This leads to misleading averages of 2D and 3D mIoU results.
>
> **A7**: Initially, xMUDA used "2D+3D" to denote the final results, but we found that "2D+3D" might be misleading as it implies an average of 2D and 3D. Therefore, we follow MM2D3D and VFMSeg by using "Avg" to denote the final result. Perhaps "Ensemble Result" (short for "Ens") would be more appropriate. Notably, in all your answers, we have changed "Avg" to "Ens".

---

> > ### Comment · Reviewer_4rFB · 2024-08-11
> >
> > Q2 mentioned that the design of LST is similar to Rein [1]. Please explain the differences between them.

---

> > > ### Author Response · Authors · 2024-08-12
> > > **Reply to Q2**
> > >
> > > **Q**: Q2 mentioned that the design of LST is similar to Rein [1]. Please explain the differences between them.
> > >
> > > **A**: Due to character limit constraints, we regret to have overlooked this question. The following is an explanation of the problem:
> > >
> > > **Similarity**: Both of them use the learnable tokens and calculate affinity matrices between visual inputs and tokens.
> > >
> > > **Difference**: Structure. Rein generates a low-rank token sequence, while our method additionally processes the visual features through a down-projection linear layer followed by an up-projection linear layer, which are then element-wise added to the residual-connected features.
> > >
> > > Following table shows simple experiment on "USA/Sing." scenario. Due to rebuttal time limitation, we will supplement all analysis between Rein and our method subsequently, including algorithm, diagram, and experimental results.
> > >
> > > | Task | Method | Visual Backbone | 2D | 3D | Avg |
> > > |------|:-----:|:-----:|:-----:|:-----:|:-----:|
> > > | DG | Use Rein | CLIP: ViT-B | 63.3 | 64.6 | 71.2 |
> > > | DG | Use LST | CLIP: ViT-B | 63.8 | 64.7 | 71.5 |

---

### Official Review · Reviewer_KKSy · 2024-07-13

**Soundness:** 3
**Presentation:** 2
**Contribution:** 2
**Rating:** 4
**Confidence:** 4

**Summary:**

This paper proposes a cross-domain 3D semantic segmentation model which utilizes off-the-shelf visual foundation models to boost the adaptability and generalizability. Two key designs are described to help the cross-domain task, e.g., visual prompt learning and deep query learning. Extensive experiments have been reported in the paper to illustrate the strength.

**Strengths:**

1. Experiments demonstrate that the proposed model indeed outperforms compared to baseline methods.
2. Appropriately freezing and finetuning strategy seems reasonable.
3. Consider prompt tuning in domain adaptation task is novel, as it can exploit the capability of VFM by prompt engineering.

**Weaknesses:**

1. The improvement in both quantitative and qualitative results does not seem to be strong enough.
2. Does not analyze the reasons behind the SOTA performance, especially compared to models like VFMSeg which also exploit the power of VFM. I hope more detailed comparison and insight can be given.
3. The motivation is not clear enough. Though the paper points out two natural questions when considering using VFM, some works have already used VFM to help with segmentation. Then the motivation should include how to improve upon these works.
4. As the model uses CLIP as the backbone, why not try a more advanced model?
5. The writing needs to be improved. For example, in line 29, the last sentence is not finished. Line 37, DG3SS does not require accessing target domain data, so the logic here is not convincing. Line 42, what do you mean by “source-domain data discrimination power”? what is the relation to your method? Line 53, what is “spatial-temporal synchronized prompt space”? Line 65, what is “pre-existing target awareness”?
Some detailed explanations could be seen in the Questions.

**Questions:**

1. I wonder about the motivation behind exploring the segmentation task in DA. As the paper has mentioned some foundation models like SAM, I want to ask that is it still meaningful to research the segmentation task in DA. A foundational segmentation model like SAM seems to be general enough (What's SAM's performance on the datasets mentioned in the paper?). Even if it's not powerful enough, for segmentation which is a kind of "high-level" task, it seems to be easier to directly improve the foundation model to achieve a real general segmentation model without any adaption.
2.  Other works like VFMSeg produce more accurate pseudo-labels to help the training, but in the limitation mentioned in this paper, producing pseudo-labels cannot achieve many performance improvements. I wonder what is the difference here.

**Limitations:**

As mentioned in the limitation of the paper, not being able to exploit the ability to produce accurate pseudo labels of VFM would constrain the capacity of the model when it comes to a larger scale.

---

> ### Author Rebuttal · Authors · 2024-08-07
>
> Dear Reviewer KKSy,
>
> We thank you for the precious review time and valuable comments. We have provided corresponding responses and results, which we believe have covered your concerns.
>
> We hope to further discuss with you whether your concerns have been addressed or not. If you still have any unclear parts of our work, please let us know. Thanks.
>
> Best,
> Authors
>
> **Q1**: The improvement does not seem to be strong enough.
>
> **A1**: Our method enhances the performance of both DG3SS and DA3SS. Notably, all loss functions are the same as those in xMUDA, and no typical data augmentation or style transfer methods are introduced to address domain shift issues. The following table shows that UniDSeg, when combined with cross-domain distillation loss from Dual-Cross, achieves best performance compared to SOTA MM2D3D. We believe that more cross-domain learning constraints can provide better model.
> | DA3SS Method | 2D | 3D | Avg |
> |-|:-:|:-:|:-:|
> | xMUDA | 55.5 | 69.2 | 67.4 |
> | Dual-Cross | 58.5 | 69.7 | 68.0 |
> | MM2D3D | 70.5 | 70.2 | 72.1 |
> | UniDSeg | 63.2 | 71.2 | 71.2 |
> | **UniDSeg + Dual-Cross** | 64.5 | 71.6 | **72.5 (+1.3)** |
>
> When combining with MM2D3D, a parallel reinitialized encoder is added to the 2D backbone to process the sparse depth. We have shown the comparisons in Table 8. On Sing./USA scenario, compared to (a), (b) not only requires 0.6% fine-tuning parameters of the 2D backbone but also achieves superior segmentation performance, making it flexible to expand to various 3D semantic segmentation tasks with multi-modal learning. For convenience, we reshow the table below:
> | Role of Depth | Params | 2D | 3D | Avg |
> |-|:-:|:-:|:-:|:-:|
> | (a) Deep Encoding | 86.9M | 66.1 | 67.7 | 73.0 |
> | (b) Point-level Prompts | 0.48M | 67.8 | 67.9 | 73.8 |
>
> **Q2**: Analyze the reasons behind the SOTA performance.
>
> **A2**: VFMSeg essentially utilizes SAM to generate instance masks for both the source and target images, applying mixing augmentation to the source and target domain data. This is a typical and useful method used to address domain shift in DA3SS. In contrast, our proposed UniDSeg exploits the image encoder of VFMs (e.g., CLIP and SAM) to extract robust visual prior information. UniDSeg proposes a learnable-parameter-inspired mechanism to the off-the-shelf image encoder of VFMs, which maximally preserves pre-existing target awareness in VFMs to further enhance its generalizability.
>
> **Q3**: The motivation is not clear enough.
>
> **A3**: Thank you for your suggestion. The motivation of this work is to study a **universal framework based on VFMs** to enhance the generalizability and adaptability of cross-domain 3D semantic segmentation, demonstrating the effectiveness of the visual foundation model priors. Different from VFMSeg, which utilize SAM to generate instance masks for both the source and target images, applying mixing augmentation to the source and target domain data. Our proposed UniDSeg is groundbreaking in introducing the **depth-guided prompt-tuning** concept into the image encoder of VFMs, which can be applied to various downstream tasks, including DA3SS, DG3SS, SFDA3SS, and fully-supervised 3SS.
>
> **Q4**: Why not try a more advanced model?
>
> **A4**: We supplement the experimental results for DA3SS and DG3SS under SAM: ViT-L. As shown in the table below, SAM-based UniDSeg exhibit better performance on "USA/Sing." scenario.
> | Task | Method | Visual Backbone | 2D | 3D | Avg |
> |-|:-:|:-:|:-:|:-:|:-:|
> | DG | UniDSeg | CLIP: ViT-L | 66.5 | 64.5 | 72.3 |
> | | | **SAM: ViT-L** | 66.8 | 64.7 | **72.6 (+0.3)** |
> | DA | UniDSeg | CLIP: ViT-L | 67.2 | 67.6 | 72.9 |
> | | | **SAM: ViT-L** | 67.8 | 68.8 | **73.3 (+0.4)** |
>
> **Q5**: Some detailed explanations could be seen.
>
> **A5**: (1) "source-domain data discrimination power" refers to the scenario in the DG3SS task where learning is conducted solely access to source domain data, enabling the model to develop the ability to discriminate domain-specific and domain-agnostic features; (2) Spatial: introducing spatial information (depth) into the prompt space to supplement 2D images; Temporal: since most scenes are captured smoothly, we learn the temporal invariance from the temporal correlations between the adjacent frames; (3) "pre-existing target awareness": In fine-tuning stage, the weights of VFMs are conventionally used to initialize source models and subsequently discarded. However, VFMs have diverse features important for generalization, and finetuning on source data can overfit to source distribution and potentially lose pre-existing target information.
>
> **Q6**: What's SAM's performance on the datasets?
>
> **A6**: In Table 2 of our paper, the experimental results demonstrate that, whether freezing or fine-tuning the image encoder of the VFM, it cannot directly solve the domain shift issue present in DG3SS and DA3SS. We supplement the experimental comparison between fine-tuning and our method, implemented on SAM image encoder serving as a visual backbone. The motivation please refer to **A3**. For other high-level tasks, they are not the focus of this work.
> | Task | Visual Backbone | Strategy | 2D | 3D | Avg |
> |-|:-:|:-:|:-:|:-:|:-:|
> | DG | SAM: ViT-L | Fine-tuning | 65.9 | 64.3 | 70.8 |
> | | | Ours | 66.8 | 64.7 | 72.6 |
> | DA | SAM: ViT-L | Fine-tuning | 66.5 | 67.9 | 71.4 |
> | | | Ours | 67.8 | 68.8 | 73.3 |
>
> **Q7**: Pseudo-label difference and improvement compared to VFMSeg.
>
> **A7**: Based on the VFMSeg experimental results, this method did not provide helpful pseudo-labels in "Day/Night" scenario, and there was even a decrease (-0.4) in performance. VFMSeg uses the averaging the probabilistic prediction of pretrained 2D network and SEEM to generate pixel-wise pseudo-labels, while our method follows the common practice adopted by most DA3SS methods in using offline 2D pseudo-label.
> | DA3SS Method | 2D mIoU |
> |-|:-:|
> | xMUDA | 55.5 |
> | xMUDA+PL | 57.5 |
> | VFMSeg+PL | 57.1 (-0.4) |
> | UniDSeg | 63.2 |
> | **UniDSeg+PL** | 64.5 (+1.3) |

---

### Official Review · Reviewer_DeKz · 2024-07-14

**Soundness:** 4
**Presentation:** 3
**Contribution:** 3
**Rating:** 7
**Confidence:** 3

**Summary:**

The manuscript proposes a universal method with the help of off-the-shelf Visual Foundation Models (VFMs) to boost the adaptability and generalizability of cross-domain 3D semantic segmentation, dubbed UniDSeg. The proposed method focus on learning visual prompt for 3D-2D transitional prior and deep query.

**Strengths:**

1.This work is the first to introduce prompt-tuning concept into the universal model for DG3SS and DA3SS.
2.The authors propose a learnable-parameter-inspired mechanism to the off-the-shelf VFMs for enhancing generalizability of VFMs.
3.The proposed method achieves state-of-the-art results on DG3SS and DA3SS tasks.

**Weaknesses:**

1.What does Samp. in Figure 1 means, the corresponding explanation is not provided in the manuscript.
2.Why choose CLIP as 2D backbone, since there is no language information for the task, other Visual Foundation Models (e.g. Swin-Transformer, SAM, …) seem more suitable for the 2D information extraction.
3.In lines 148-152, the author mention “The main insight of UniDSeg is to provide a universal framework that enhances the adaptability and generalizability of cross-domain 3D semantic segmentation.” Though SparseConvNet is examined as 3D network, it would be more convincing as “universal” if authors could provide results utilizing other 3D backbones.

**Questions:**

Please refer to the Weaknesses Section.

**Limitations:**

The authors have clearly discussed the potential limitations of this work.

---

> ### Author Rebuttal · Authors · 2024-08-07
>
> Dear Reviewer DeKz,
>
> We thank you for the precious review time and valuable comments. We have provided corresponding responses and results, which we believe have covered your concerns.
>
> We hope to further discuss with you whether your concerns have been addressed or not. If you still have any unclear parts of our work, please let us know. Thanks.
>
> Best,
> Authors
>
> **Q1**:  What does Samp. in Figure 1 means, the corresponding explanation is not provided in the manuscript.
>
> **A1**: "Samp." means sampling of 2D features. Only the points falling into the intersected field of view are geometrically associated with multi-modal data (i.e., 3D-to-2D projection). After "Samp." operation, we can obtain point-wise 2D features for cross-modal learning with 3D features.
>
> **Q2**: Why choose CLIP as 2D backbone, since there is no language information for the task, other Visual Foundation Models (e.g. Swin Transformer, SAM, …) seem more suitable for the 2D information extraction.
>
> **A2**: Thank you for your reminder. The motivation of this work is to study a **universal framework based on VFMs** to enhance the generalizability and adaptability of cross-domain 3D semantic segmentation, demonstrating the effectiveness of the visual foundation model priors. To this end, we utilize the off-the-shelf image encoder of VFMs as our visual backbone. UniDSeg **can be applied to various downstream tasks, including DA3SS, DG3SS, SFDA3SS, and fully-supervised 3SS**. Following your suggestion, we have supplemented the experimental results for DA3SS and DG3SS under another visual backbone, SAM: ViT-L. As shown in the table below, SAM-based UniDSeg exhibit better performance on "USA/Sing." scenario.
> | Task | Method | Visual Backbone | 2D | 3D | Avg |
> |-|:-:|:-:|:-:|:-:|:-:|
> | DG | UniDSeg | CLIP: ViT-L | 66.5 | 64.5 | 72.3 |
> | | | **SAM: ViT-L** | **66.8** | **64.7** | **72.6 (+0.3)** |
> | DA | UniDSeg | CLIP: ViT-L | 67.2 | 67.6 | 72.9 |
> | | | **SAM: ViT-L** | **67.8** | **68.8** | **73.3 (+0.4)** |
>
> Furthermore, we supplement the experimental comparison between fine-tuning and our method, implemented on SAM image encoder serving as a visual backbone. Due to rebuttal time limitation, we will supplement all SAM-based experimental results subsequently.
> | Task | Visual Backbone | Strategy | 2D | 3D | Avg |
> |-|:-:|:-:|:-:|:-:|:-:|
> | DG | SAM: ViT-L | Fine-tuning | 65.9 | 64.3 | 70.8 |
> | | | Ours | 66.8 | 64.7 | 72.6 |
> | DA | SAM: ViT-L | Fine-tuning | 66.5 | 67.9 | 71.4 |
> | | | Ours | 67.8 | 68.8 | 73.3 |
>
> **Q3**: Though SparseConvNet is examined as 3D network, it would be more convincing as "universal" if authors could provide results utilizing other 3D backbones.
>
> **A3**: Thank you for your suggestion. We have supplemented the DA3SS experimental results for xMUDA and our proposed UniDSeg under another 3D backbone network, MinkowskiNet. Due to rebuttal time limitation, we will supplement all MinkowskiNet-based experimental results subsequently.
> | 3D Backbone | DA3SS Method | 2D | 3D | Avg |
> |-|:-:|:-:|:-:|:-:|
> | SparseConvNet | xMUDA | 64.4 | 63.2 | 69.4 |
> | | UniDSeg | 67.2 | 67.6 | 72.9 |
> | MinkowskiNet | xMUDA | 65.9 | 64.0 | 69.7 |
> | | UniDSeg | 67.5 | 68.6 | 73.1 |

---

> ### Comment · Reviewer_DeKz · 2024-08-14
> **Response to Authors**
>
> The authors' response has clearly addressed my concerns. After checking the peer review comments and the author's responses, I decided to raise the given score for this work.

---

### Decision · Program_Chairs · 2024-09-25

**Decision:**

Accept (poster)

**Comment:**

This paper receives ratings with 1 accept, 1 borderline accept, and 2 borderline rejects. Most reviewers like the idea of bringing prompt tuning in to the domains of DG3SS and DA3SS and the extensive experiments to verify the proposed method. The main concerns raised by the reviewers including the use of CLIP as the backbone, novelty compared to existing 3SS methods that utilize VFMs, and marginal improvement against existing methods. The authors provide very detailed rebuttal to address the raised concerns. After reading the paper, reviews, and rebuttal, AC finds that the raised concerns are mostly addressed by the rebuttal with some nice ablation studies, and the improvement against the existing method is non-trivial. Given the novelty of the proposed methods, AC believes that the paper is ready for publication. The authors are encouraged to make the above mentioned necessary changes to the best of their ability. We congratulate the authors on the acceptance of their paper!